# Isolation and characterization of plant-derived exosome-like nanoparticles from *Carica papaya* L. fruit and their potential as anti-inflammatory agent

Iriawati Iriawati⑩, Safira Vitasasti, Fatimah Nur Azmi Rahmadian, Anggraini Barlian⑩*

School of Life Sciences and Technology, Institut Teknologi Bandung, Bandung, Indonesia

* aang@sith.itb.ac.id

**Data Availability Statement:** All relevant data are within the manuscript and its Supporting Information files.

## Abstract

Inflammation is an immune system response that identifies and eliminates foreign material. However, excessive and persistent inflammation could disrupt the healing process. Plant-derived exosome-like nanoparticles (PDENs) are a promising candidate for therapeutic application because they are safe, biodegradable and biocompatible. In this study, papaya PDENs were isolated by a PEG6000-based method and characterized by dynamic light scattering (DLS), transmission Electron Microscopy (TEM), bicinchoninic acid (BCA) assay method, GC-MS analysis, total phenolic content (TPC) analysis, and 2,2-diphenyl-1-picryl-hydrazyl (DPPH) assay. For the in vitro test, we conducted internalization analysis, toxicity assessment, determination of nitrite concentration, and assessed the expression of inflammatory cytokine genes using qRT-PCR in RAW 264.7 cells. For the in vivo test, inflammation was induced by caudal fin amputation followed by analysis of macrophage and neutrophil migration in zebrafish (*Danio rerio*) larvae. The result showed that papaya PDENs can be well isolated using the optimized differential centrifugation method with the addition of 30 ppm pectolyase, 15% PEG, and 0.2 M NaCl, which exhibited cup-shaped and spherical morphological structure with an average diameter of 168.8±9.62 nm. The papaya PDENs storage is stable in aquabidest and 25 mM trehalose solution at -20˚C until the fourth week. TPC estimation of all papaya PDENs ages did not show a significant change, while the DPPH test exhibited a significant change in the second week. The major compounds contained in Papaya PDENs is 2,3-dihydro-3,5-dihydroxy-6-methyl-4H-pyran-4-one (DDMP). Papaya PDENs can be internalized and is non-cytotoxic to RAW 264.7 cells. Moreover, LPS-induced RAW 264.7 cells treated with papaya PDENs showed a decrease in NO production and downregulation mRNA expression of pro-inflammatory cytokine genes (IL-1B and IL-6) and an upregulation in mRNA expression of anti-inflammatory cytokine gene (IL-10). In addition, in vivo tests conducted on zebrafish treated with PDENs papaya showed inhibition of macrophage and neutrophil cell migration. These findings suggest that PDENs papaya possesses anti-inflammatory properties.

**Funding:** This Research was funded by Penelitian, Pengabdian Kepada Masyarakat dan Inovasi (PPMI) Sekolah Ilmu dan Teknologi Hayati, Institut Teknologi Bandung Fund under Grant (52.A/IT1.CII/ 5K-PL/2022). The funders had no role in study design, data collection and analysis, decision to publish, or preparation of the manuscript.

**Competing interests:** The authors have declared that no competing interests exist.

## Introduction

Recently, exosome nanotechnology has gained popularity as a therapeutic strategy. In this study, we used exosomes derived from plants which are typically 50–500 nm in size and referred to them as plant-derived exosome-like nanoparticles (PDENs) [1–3]. PDENs serves as a carrier for diverse cargoes, such as microRNA, proteins, lipids, vitamins, and metabolites [4]. The cargoes are encapsulated to protect it from degradation and metabolism [5]. PDENs can also be delivered across species, and it has an ability to manipulate the functions of specific cells within targeted tissues, thereby optimizing its potential for nano-delivery treatments [6, 7].

Papaya fruit (*Carica papaya* L.) is commonly used in traditional herbal medicine for wound healing due to the presence of proteolytic enzymes. Additionally, papaya contains a variety of phytochemicals and exhibits antimicrobial, antioxidant, and anti-inflammatory properties that prove useful in the treatment of chronic skin wounds. These properties can significantly enhance the wound healing process and protect the tissue from oxidative damage [6–8]. However, papaya fruit contains high levels of pectin, which is indicated by high viscosity, making it difficult to extract due to the bond between the juice and the pulp in a jelly-like form [9].

Experimental evidence indicates that papaya extract could downregulate pro-inflammatory genes, including interleukin 4 (IL-4), interleukin 5 (IL-5), tumor necrosis factor α (TNF-α), nuclear factor-κB (NF-κB), and inducible nitric oxide synthase (iNOS) [10]. Another study demonstrated that papaya extract can enhance the regulation of the interleukin 10 gene (IL-10) and reduce the expression of the interleukin 6 (IL-6) and TNF-α genes [11]. PDENs is expected to possess therapeutic properties that are similar to its origin [12], therefore PDENs isolated from papaya fruit has the potential to act as an anti-inflammatory agent due to its similar therapeutic properties to the source.

Inflammation is part of the body's innate immune response (innate immunity) when exposed to stimuli from foreign agents [13]. The macrophages serve as the first line of defense by releasing cellular signaling molecules, pro-inflammatory cytokines (TNF-α, IFN-γ, IL-1β, and IL-6), and inflammatory mediators (NO, PGE2, and COX-2) [14, 15]. Continuous inflammation can lead to chronic inflammatory diseases like cancer, diabetes, and heart disease [16]. Using anti-inflammatory agents is one approach to managing chronic inflammation.

The use of anti-inflammatory agents like non-steroidal anti-inflammatory drugs (NSAIDs) and glucocorticoids holds promise for treating chronic inflammation-associated diseases (CID). However, prolonged administration of these drugs may result in liver, kidney, and intestinal mucosal damage, limiting their use [17–19]. Thus, the development of effective and safe therapeutic agents to manage severe inflammation is crucial.

PDENs has exciting potential as a therapeutic agent or drug delivery vehicle. It exhibits excellent biocompatibility and can target specific tissues through precise endocytosis mechanisms, thus reducing off-target effects and broadening the scope of drug therapy [2]. Studies have demonstrated that PDENs derived from ginger can induce anti-inflammatory cytokines, suggesting its potential as an anti-inflammatory agent. Twenty-seven microRNAs were detected, primarily involved in regulating inflammation and cancer pathways [20]. Studies have shown that broccoli PDENs can inhibit monocytes at the site of inflammation and thus reducing levels of pro-inflammatory cytokines. During the in vivo tests, an increase in anti-inflammatory cytokines was observed, which prevented both acute and chronic colitis and minimized side effects by regulating the activation of AMP-activated protein kinase (AMPK) [21]. Another study demonstrated that Mandarin Orange PDENs, and *Aloe saponaria* PDENs can lower the expression of genes related to pro-inflammatory cytokines in RAW 264.7 cells [22, 23].

Zebrafish (*Danio rerio*) belongs to the Cyprinidae family and is a teleost fish. Researchers commonly use zebrafish as a model system to investigate various pathologies due to their exceptional regenerative capabilities, which are unparalleled by mammals in their ability to quickly repair or replace damaged cells [24, 25]. Furthermore, zebrafish has demonstrated several advantages over other animal species and may serve as a substitute for rodents or provide supplementary information when used as a model for studying inflammatory diseases. Research shows that zebrafish possess an immune system similar to that of humans, making them a suitable model for such studies [26–28]. In addition, zebrafish have the advantage of low breeding costs, transparent embryos and larvae that allow the observation of specific cells, and permeability to small compounds [29, 30].

This is the first study on papaya PDENs that has been conducted to date. Our goal was to optimize the isolation process of papaya PDENs, characterize, evaluate the storage stability of papaya PDENs, and analyze its anti-inflammatory activity both in vitro using RAW 264.7 cells and in vivo using zebrafish. Our findings suggest the potential of papaya PDENs as an anti-inflammatory agent.

## Materials and methods

### Optimization of isolation of papaya exosome-like nanoparticles (papaya PDENs) by differential centrifugation and PEG method

PDENs isolation from papaya was performed according to Kalarikkal *et al.* [31] by using a polyethylene glycol-6000 (PEG6000) (Himedia, India) based method with modifications. Papaya obtained from the local market with two different ages, raw and ripe papaya, was peeled and ground using a juicer. The juice was then filtered thoroughly using two types of nylon mesh (100 μm then 40 μm). The high pectin content in papaya made the extraction process difficult to carry out without any modification process to the filtrate. The obtained filtrate was then incubated with various concentrations of pectolyase (Pectolyase Y-23 produced by Seishin Pharmaceutical Co., LTD., Tokyo, Japan); 0, 10, 20, 30, 40, 50, 75, and 100 ppm, and kept on a shaker at 120 rpm for 90 min at room temperature using OHAUS Light Duty Orbital Shakers. The juice was then mixed with aquabidest (Ikapharmindo, Indonesia) in different ratios; 1:4, 1:2, 1:1.5, and 1:1 v/v. In order to eliminate impurities such as fibers, cell debris, extracellular matrix, and organic/inorganic compounds, the mixture was subjected to serial centrifugation steps. Specifically, the mixture was centrifuged at 2000g for 10 min, 6000g for 20 min, and finally at 10000g for 40 min, all at 4°C. The supernatant was mixed with PEG6000 to reach several final concentrations; 5%, 10%, and 15% (v/v) with the additional 0.1 M, 0.2 M, and 0.5 M NaCl (Sigma-Aldrich, USA). The mixtures were then incubated overnight at 4°C. After centrifugation at 8000 g for 30 min at 4°C, the excess supernatant was removed by inverting the tube on a piece of tissue paper. The pellet was resuspended in aquabidest, 25 mM trehalose was subsequently added and filtered through a 0.45 μm then 0.22 μm Millipore filter (Minisart Sartorius, German).

### Protein quantification

The protein concentration (μg/ml) of Papaya PDENs was measured by the bicinchoninic acid assay according to the BCA Protein Assay Kit manufacturer's protocols (Thermo Fisher Scientific, USA). Briefly, the BSA standard curve was prepared from the dilution of one BSA ampule into several clean vials, and the BCA working reagent was prepared by mixing BCA Reagent A with BCA Reagent B (50:1). All samples in triplicates were measured by microplate reader at

595 nm absorbance. The BSA standard curve was used to calculate the concentration of the Papaya PDENs protein.

## Size and morphology of papaya PDENs

The physical characteristics of Papaya PDENs were analyzed by evaluating their particle size, zeta potential, and morphology. The Papaya PDENs was analyzed by dynamic light scattering (DLS) using Nano Particle Analyzer Horiba SZ-100 to measure particle size (mean diameter) and zeta potential. Each sample was measured 3 times at room temperature. Subsequently, the morphology of the Papaya PDENs was examined using TEM HT7700.

## Storage stability of papaya PDENs

The size and morphology of Papaya PDENs were analyzed to assess their shelf life and storage temperature. Papaya PDENs were subjected to different storage durations of 0, 1, 2, 3, and 4 weeks with different temperatures of 4˚C and −20˚C. The storage stability of the Papaya PDENs was analyzed by measuring their mean diameter using dynamic light scattering (DLS) at each storage duration and temperature [32]. Afterward, the morphology of the different ages of papaya PDENs was inspected using TEM HT7700.

## Gas chromatography-mass spectrometry analysis

GC-MS analysis was conducted in the Regional Health Laboratorium of Jakarta, Indonesia, using Agilent Technologies 7890 gas chromatograph with autosampler and 5975 mass selective detector and chemstation data system. For GC-MS analysis, the papaya PDENs pellets were mixed with 5 mL of absolute methanol (Merck, Germany) until they became a homogenous solution. The mixtures (5 µl) were separated using an HP Ultra 2 capillary column (30 m length, 0.20 mm internal diameter, 0.11 µm film). The GC oven conditions were as follows: initial temperature 80˚C hold for 1 minute, rising at 3˚C/min to 150˚C hold for 1 minute, and rising 20˚C/min to 280˚C, hold for 26 minutes. Helium was used as a carrier gas at a 1.2 mL/minute constant flow. The transfer line and ion source temperature were 250˚C. Electron impact ionization was used at 70 eV. Compounds were identified through mass spectral matching using the NIST database.

## Total polyphenolic content (TPC) estimation of papaya PDENs

Total polyphenolics from papaya PDENs were purified by methanol extraction. Briefly, at room temperature, 20 µl of papaya PDENs were mixed with 100 µl of absolute methanol, vortexed, and incubated for 10 minutes. The supernatant fraction was used to estimate TPC using the modified protocol [31], after centrifugation at 10000g for 5 minutes. In brief, the methanol extract sample was mixed with 200 µl of Folin-Ciocalteu reagent (Merck, Germany) and vortexed. After the addition of 2 mL of 7% sodium Carbonate (Sigma-Aldrich, USA), samples were dark incubated for 30 minutes at room temperature. Samples were transferred to cuvette and the blue color developed was measured using a spectrophotometer at 765 nm wavelength, with absolute methanol as a blank. Gallic acid (Sigma-Aldrich, USA) was used to generate a standard curve to represent TPC values as a gallic acid equivalent per gram of sample.

## 1,1-diphenyl-2-picrylhydrazyl (DPPH) assay for antioxidant activity of papaya PDENs

The free radical scavenging activity of papaya PDENs was conducted using an adopted protocol from Kalarikkal *et al.* [31]. Briefly, 7.89 mg DPPH reagent (Sigma-Aldrich, USA) was

dissolved in 100 mL absolute methanol to reach a final concentration of 0.2 mM. The solution was incubated in the dark at room temperature for 2 hours until achieve the stable colometric absorbance. Purified phytochemicals from papaya PDENs are obtained by methanol extraction as mentioned before. 900 μl of DPPH reagent was mixed with 100 μl methanol extract and was kept in the dark for 30 minutes at room temperature. Absorbance was measured at 517 nm wavelength using a spectrophotometer, with DPPH reagent alone serving as a control, and absolute methanol as a blank. The DPPH antioxidant activity was calculated using the radical scavenging formula.

$$Radical\ scavenging(\%) = \left[\frac{(A)control - (A)sample}{(A)control}\right] x\ 100$$

## RAW 264.7 cell lines

RAW 264.7 cell line (CL-0190) (Elabscience, China) were cultured in DMEM supplement with 10% (v/v) fetal bovine serum and 1% (v/v) antibiotic-antimycotic solution (Gibco, USA) at 37°C in 5% $CO_2$ and 80% humidity. After the cells reached 80%–90% confluency, they were harvested using cell scraper (Biologix, USA) and subcultured.

## Uptake of papaya PDENs by RAW 264.7 cells

In order to confirm the cellular uptake of PDENs, PKH67 green dye (Sigma-Aldrich, USA) was utilized to label Papaya PDENs at room temperature, followed by a centrifuge of 20.000g at 4°C for 1h. The supernatant was removed and washed with ddH2O, followed by a centrifuge of 20.000g at 4°C for 15 min twice. Free PKH67 dye was removed by filtration using a syringe filter. RAW 264.7 cells were first seeded on sterile cover slips placed in a 6-well plate at a density of $1\times10^5$ cells/well and cultured in a complete medium for 24 hours prior to labeling. Afterward, each coverslip was washed with PBS (Sigma-Aldrich, USA) before labeled Papaya PDENs was added to wells and incubated for 2, 4, 6, and 24 h at 37°C. Afterward, the cells on the coverslips were fixed with 4% paraformaldehyde (Sigma-Aldrich, USA) and nuclear stained with DAPI (Thermo Fisher Scientific, USA). Intracellular uptake of Papaya PDENs was analyzed using a confocal laser scanning microscope Olympus FV-1200 [31, 33].

## Cell viability assay

RAW 264.7 cells were seeded in 96-well plates at a density of $1 \times 10^4$ cells/well. 24 h after seeding, cells were washed and incubated with 5, 10, 20, 50, and 100 μg/ml Papaya PDENs for 24 h. To check the cell viability, 10 μL MTT 5 mg/mL (3-[4,5-Dimethylthiazol-2-yl]-2,5-diphenylte-trazolium bromide) (Thermo Fisher Scientific, USA) was added to each well and incubated at 37°C for 4 hours. The formed formazan crystals were dissolved using 100 μL/well of DMSO [34]. Optical density (OD) is measured at a wavelength of 595 nm using a microplate reader Bio-Rad iMark and the percentage of viable cells was calculated as:

$$Type\ \%\ \text{viable cells} = \left(\frac{\text{OD treatments}}{\text{OD untreated control}}\right) \times 100$$

## Nitric oxide assay

RAW 264.7 cells were cultured in 96-well plates, with a density of $1 \times 10^4$ cells per well and incubated for 24 hours. Following this, they were exposed to various concentrations of PDENs (0, 5, 10, and 20 μg/ml) and dexamethasone (1 μg/ml) (Sigma-Aldrich, USA) for 24 hours.

Thereafter, the cells were treated with LPS (50 ng/mL) (Sigma-Aldrich, USA) for an additional 6 hours. To measure the nitric oxide in the culture media, we utilized the Protocol from Griess Reagent Kit for Nitrite Determination (G-7921) (Thermo Fisher Scientific, USA). Briefly, 50 microliters of cell culture medium from each treatment were mixed with Griess reagent (1:1) and incubated for 15 minutes in the dark. The samples were measured at a wavelength of 540 nm using a microplate reader Bio-Rad iMark.

## Total RNA extraction and qRT-PCR

RAW 264.7 cells were cultured in 6-well plates at a density of $2 \times 10^5$ cells per well and incubated for 24 hours. Next, the cells were treated with PDENs at various concentrations (0, 5, 20, and 100 μg/mL) and dexamethasone (1 μg/ml) (Sigma-Aldrich, USA) for 24 hours. Following that, the cells were exposed to LPS (50 ng/mL) (Sigma-Aldrich, USA) for 6 hours. Total RNA was then extracted from the cultured RAW 264.7 cells using the Quick-RNA$^{TM}$ MiniPrep Plus kit as per the manufacturer's instructions (Zymo Research, USA). The quantity and purity of the extracted total RNA samples were measured using a NanoDrop Lite Spectrophotometer. Complementary DNA (cDNA) synthesis was conducted using Promega GoTaq 2-Step RT-qPCR System protocol (Promega, USA) then polymerase chain reaction (PCR) was performed in QuantStudio 1 Real-Time PCR System. Lists of the PCR primers for each gene are shown in Table 1.

## Treatment of zebrafish larvae

All animal experiments were approved by the Research Ethics Committee Universitas Padjadjaran Bandung (ID number: 800/UN6.KEP/EC/2023) and all efforts were made to minimize suffering. Fertilized embryos were obtained from pairs of adult fish by natural spawning and cultured at 28.5 ˚C in clean Petri dishes with E3 medium (5 mM NaCl, 0.17 mM KCL, 0.33 mM CaCl, and 0.33 mM MgSO4) (Sigma-Aldrich, USA). A solution of 0.003% 1-phenyl-2-thiourea (PTU) (Sigma-Aldrich, USA) was added to the 10 hpf embryos to prevent melanin formation. Afterward, 1 dpf embryos were treated with varying concentrations of Papaya PDENs (5, 10, 20 μg/mL) and dexamethasone (300 μM) for 6 h and the control group was treated with E3 medium. The treatment was conducted using a 6-well plate (30 embryos/well). The embryos were then washed three times with medium E3 and placed in new plates. Afterward, 3 dpf embryos were anesthetized with 0.04% tricaine (Sigma-Aldrich, USA) and the caudal fin was subjected to amputation. The embryos were given a recovery period of 4 hours after amputation before being stained by neutral red (NR) and sudan black (SB) [35, 36].

**Table 1. PCR primers.**

| | |
|---|---|
| **GAPDH** | |
| Forward | 5' - TGTGTCCGTCGTGGATCTGA - 3' |
| Reverse | 5' - TTGCTGTTGAAGTCGCAGGAG- 3' |
| **Interleukin-6 (IL-6)** | |
| Forward | 5'- ATCCAGTTGCCTTCTTGGGA - 3' |
| Reverse | 5'- GGTCTGTTGGGAGTGGTATCC - 3' |
| **Interleukin-1β (IL-1β)** | |
| Forward | 5' - TGCCACCTTTTGACAGTGATG - 3' |
| Reverse | 5' - ATGTGCTGCTGCGAGATTTG - 3' |
| **Interleukin-10 (IL-10)** | |
| Forward | 5' - AGACCAAGGTGTCTACAAGGC- 3' |
| Reverse | 5' - CCAAGGAGTTGTTTCCGTTAGC - 3' |

### Neutral red staining

In order to stain macrophages, the larvae at 3 dpf were incubated with 2.5 μg/mL neutral red (Sigma-Aldrich, USA) and 0.003% PTU (Sigma-Aldrich, USA) at room temperature in the dark for 45 min to 1 h. Afterward, the neutral red solution was removed and exchanged with PTU. The macrophage migration was observed using a stereo microscope. The total number of macrophage cells was counted with ImageJ Version 1.54d software [37, 38].

### Sudan black staining

In order to stain neutrophils, the working solution of Sudan Black was prepared from 0.6 g of SB powder (Sigma-Aldrich, USA) in 200 mL of ethanol absolute (Merck, USA), followed by filtration and stored at 4˚C. The buffer solution was prepared by mixing 16 g of phenol (Sigma-Aldrich, USA) in 30 mL of pure ethanol and 0.3 g of $Na_2HPO_4.12H_2O$ (Sigma-Aldrich, USA) in 100 mL of distilled water. A working solution of Sudan Black was the filtered mixture of 30 mL SB stock solution with 20 mL buffer. The larvae at 3 dpf were fixed with 4% paraformaldehyde (PFA) (Sigma-Aldrich, USA) for 2 h at room temperature, washed in PBS (Sigma-Aldrich, USA), incubated with 1 ml SB working solution at room temperature in the dark and rocked for 1 h. Afterward, the larvae were washed 3 times with 70% ethanol for 5 min. Larvae were rehydrated by washing in PBS-tween (PBS plus 0.1% Tween-20 (PBST) (Sigma-Aldrich, USA). Finally, the PSBT was removed and exchanged with PBS. The total number of neutrophil cells was counted with ImageJ 1.54d software [37, 39].

### Statistical analysis

The results are presented as a mean ± standard deviation (SD) with at least triplicate measurements. One-way ANOVA was utilized to analyze the difference between the control and test groups followed by Tukey post hoc test, and $p < 0.05$ was considered statistically significant. Statistical analyses were performed using GraphPad Prism software (GraphPad Software, Inc.).

## Result

### Optimization of isolation of papaya PDENs method

The isolation of PDENs from papaya fruit was carried out using the polyethylene glycol-based method with modifications. From the isolation results, papaya PDENs was obtained in the form of orange pellets found at the bottom of the falcon tube. The amount of pellets was seen to be the highest with the addition of 15% PEG. The presence of NaCl in the sample solution also increases the amount of pellets obtained, with the best result at 0.2 M NaCl (Fig 1A). The amount of pellets is also influenced by the concentration of PEG stock and pectolyase (Fig 1B). The optimum condition to obtain the most abundant pellets is reached by 15% PEG from 50% PEG stock combined with the additional of 40 ppm of pectolyase and 0.2 M NaCl (Fig 1C).

Dynamic light scattering (DLS) analysis of papaya PDEN showed that 15% PEG treatment has more homogeneous particle sizes than 10% PEG, ranging from 151.5–170.4 nm with an average of 160.95 ± 9.45 nm and a polydispersity index (PDI) of 0.294 ± 0.087 (Fig 2A and 2B). In addition, pectolyase treatment showed a better result at 30 ppm rather than 40 ppm, which is indicated by the range of particle sizes between 209.9–214.7 nm with an average of 212.3 ± 2.4 nm and a PDI of 0.264 ± 0.05 (Fig 2C and 2D). The best result presented at 15% PEG with the addition of 30 ppm pectolyase and 0.2 M NaCl, exhibited by the average particle size at 168.8 ± 9.62 nm and a PDI of 0.261 ± 0.045 (Fig 2E). Moreover, papaya PDENs is negatively charged with a value of -9.4 ± 0.2 mV (Fig 2F). All the data is shown in Table 2.

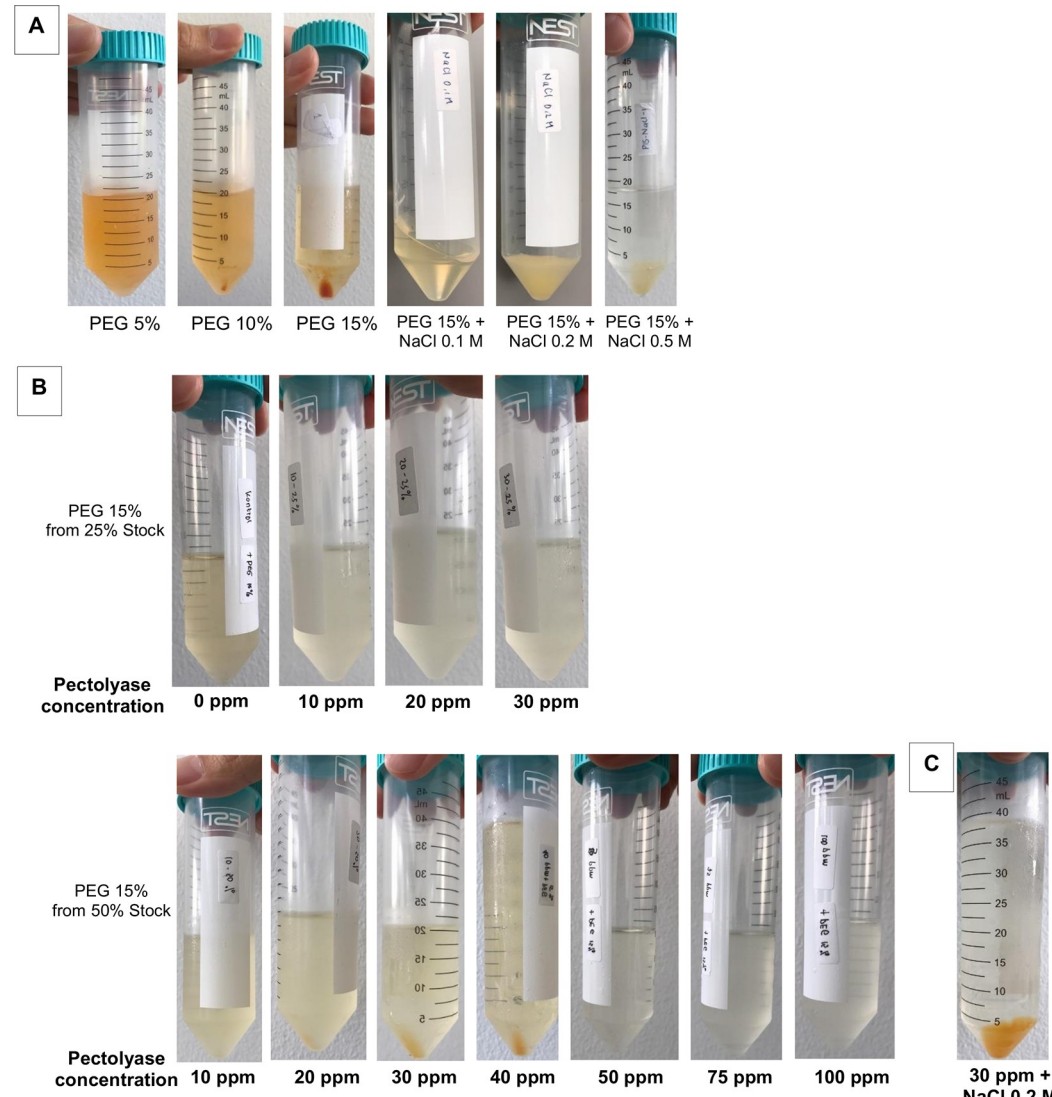

**Fig 1. Isolated pellet of papaya PDENs.** The effect of (A) Different PEG concentrations and the addition of various NaCl concentrations, and (B) Different concentrations of PEG stock and several pectolyase concentrations. (C) The optimum condition to isolate papaya PDENs.

Based on transmission electron microscopy (TEM) imaging, the morphological structure of papaya PDENs was spherical and cup-shaped with a population size of ~150–200 nm (Fig 3). The additional of NaCl 0.2 M gave a prominent result to the shape, homogeneity, and size distribution of papaya PDENs (Fig 3B) compared to sample without NaCl (Fig 3A). Furthermore, the protein concentration of papaya PDENs measured by BCA assay was in the range of 1486–1761 µg/mL.

## Storage stability of papaya PDENs

The storage stability of papaya PDENs was assessed by measuring its average diameter using the dynamic light scattering (DLS) technique at specific times: 0, 1, 2, 3, and 4 weeks and at specific storage temperatures: 4°C and −20°C. The results indicate a slight increase in the size

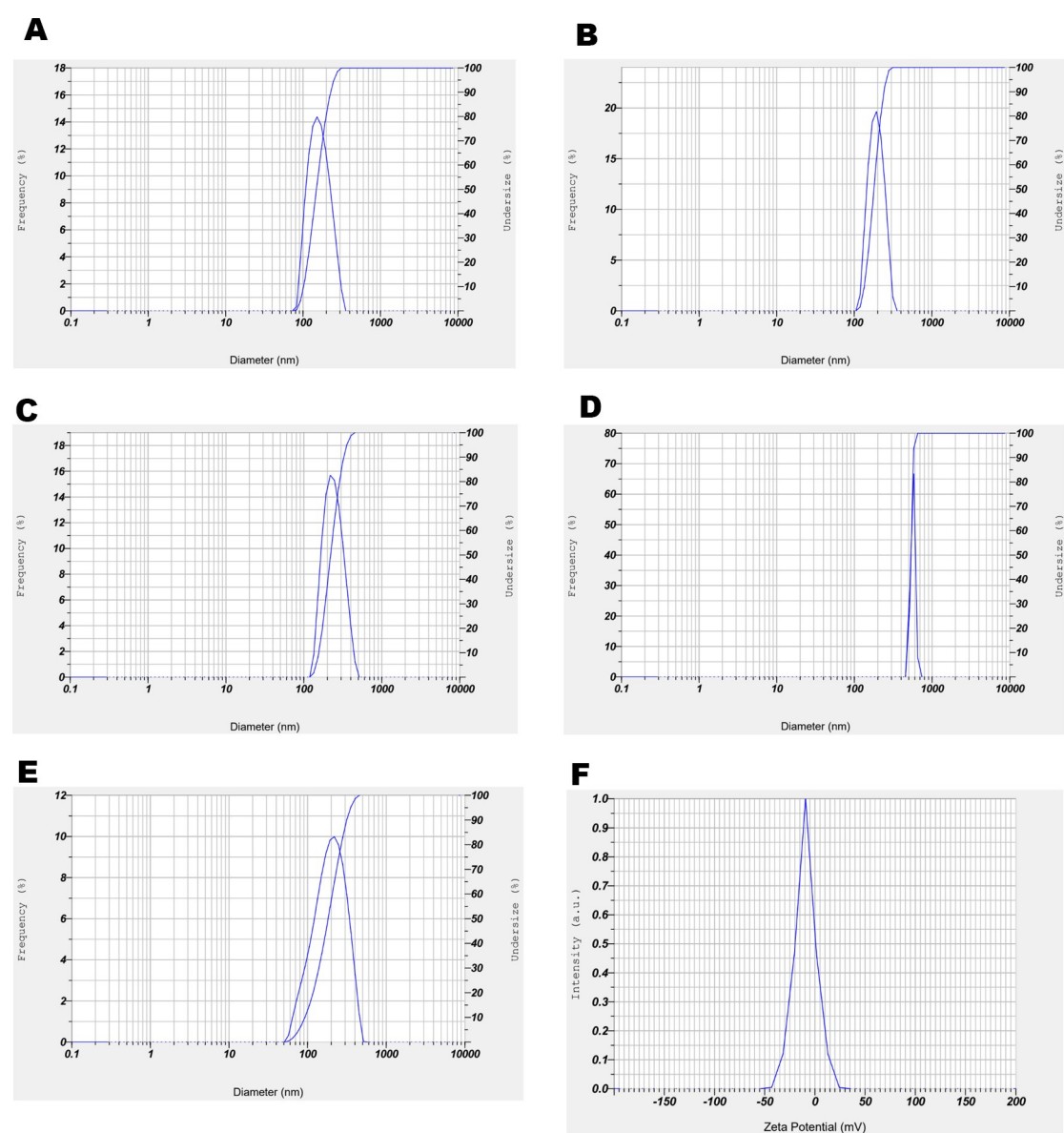

**Fig 2. The size distribution of treatment.** (A) 10% PEG6000, (B) 15% PEG6000, (C) 15% PEG6000 with 30 ppm pectolyase, (D) 15% PEG6000 with 40 ppm pectolyase, (E) 15% PEG6000 with 30 ppm pectolyase and 0.2 M NaCl. (F) Zeta potential of the papaya PDENs from sample (E).

**Table 2. The Z-average, polydispersity index, and count rate of the papaya PDENs with different treatment.**

| Treatment | Z-average (nm) | Polydispersity index | Count rate (kCPS) |
|---|---|---|---|
| PEG 10% | 119.2 ± 4.8 | 0.585 ± 0.1 | 8.5 ± 1.5 |
| PEG 15% | 160.95 ± 9.45 | 0.294 ± 0.087 | 11.5 ± 1.5 |
| PEG 15% + NaCl 0.2 M | 155.15 ± 4.95 | 1.01 ± 0.049 | 36 ± 4 |
| PEG 15% + NaCl 0.5 M | 168.6 ± 13.6 | 1.306 ± 0.058 | 18.5 ± 3.5 |
| Pectolyase 30 ppm + PEG 15% | 212.3 ± 2.4 | 0.264 ± 0.05 | 558.5 ± 85.5 |
| Pectolyase 40 ppm + PEG 15% | 425.65 ± 5.25 | 0.474 ± 0.0095 | 1172 ± 324 |
| Pectolyase 40 ppm + PEG 15% + NaCl 0.2 M | 155.3 ± 1.3 | 0.352 ± 0.014 | 624 ± 111 |

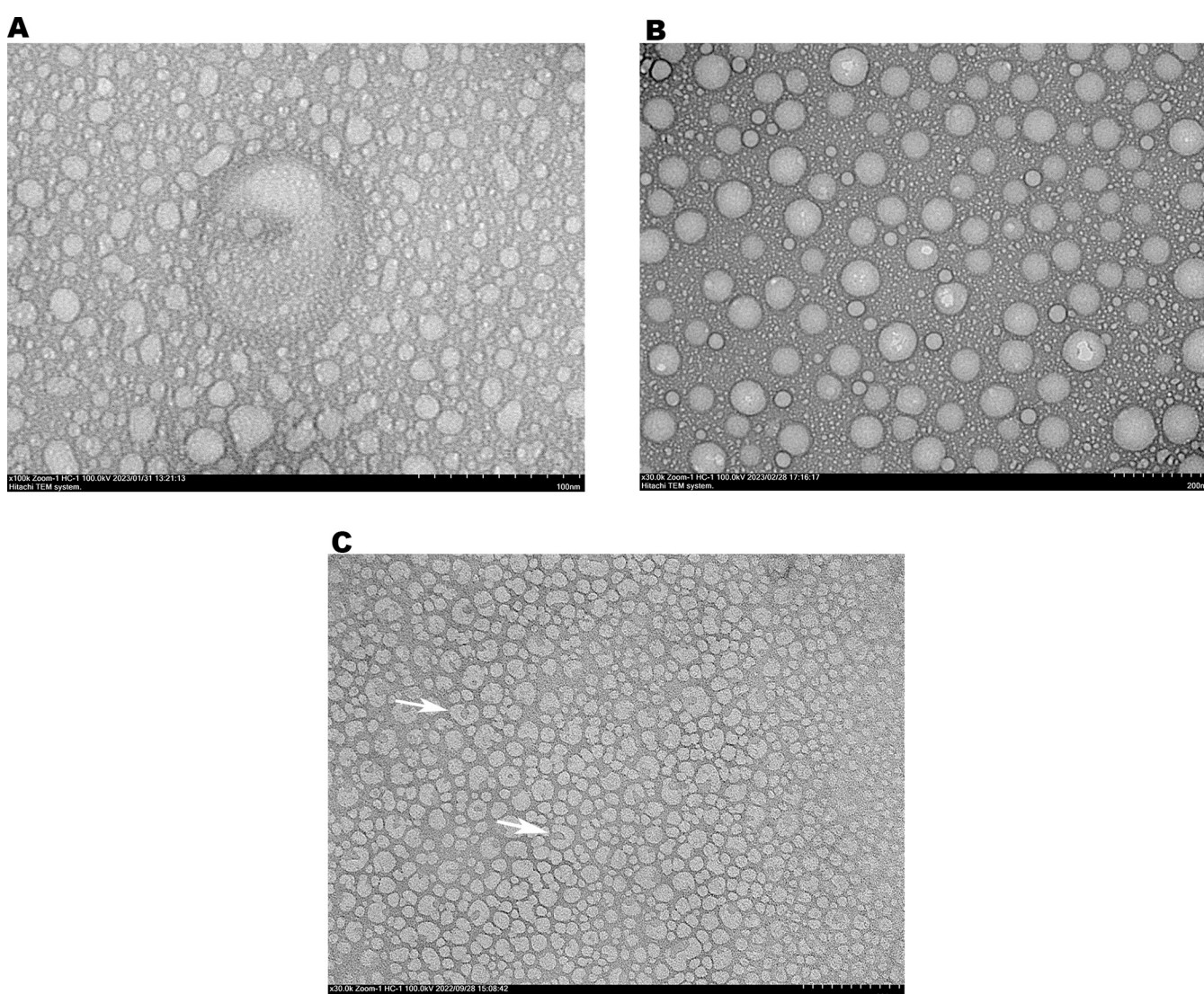

**Fig 3. The morphology of papaya PDENs were visualized by TEM with different treatment.** (A) Pectolyase 30 ppm without NaCl, and (B) Pectolyase 30 ppm with additional NaCl 0.2 M. (C) The cup-shaped (white arrow) and spherical morphological structure of papaya PDENs.

of papaya PDENs in the second week, but it remains relatively stable in the second to fourth weeks for up to 4 weeks of storage at -20˚C without any significant change (Fig 4). In contrast, storing at 4˚C resulted in a noticeable change in size in the first week, but no significant change was observed up to 4 weeks of storage. It was also observed that PDENs papaya exhibited an increasing PDI from week 1 to 4 at 4˚C (S1 Table). In addition, based on TEM imaging of samples stored at -20˚C, there were no major change in the shape and particle size distribution (Fig 5). Therefore, it can be concluded that -20˚C storage conditions are better than 4˚C in maintaining PDENs size in the range of 50–500 nm, especially $\leq$ 200 nm.

## Gas chromatography-mass spectrometry analysis

Different ages of papaya PDENs samples isolated from papaya fruit were extracted by methanol and analyzed by GC-MS to reveal metabolome profiles. GC-MS is a highly sensitive analytical technique for investigating targeted and untargeted metabolites in plants. In this study, we

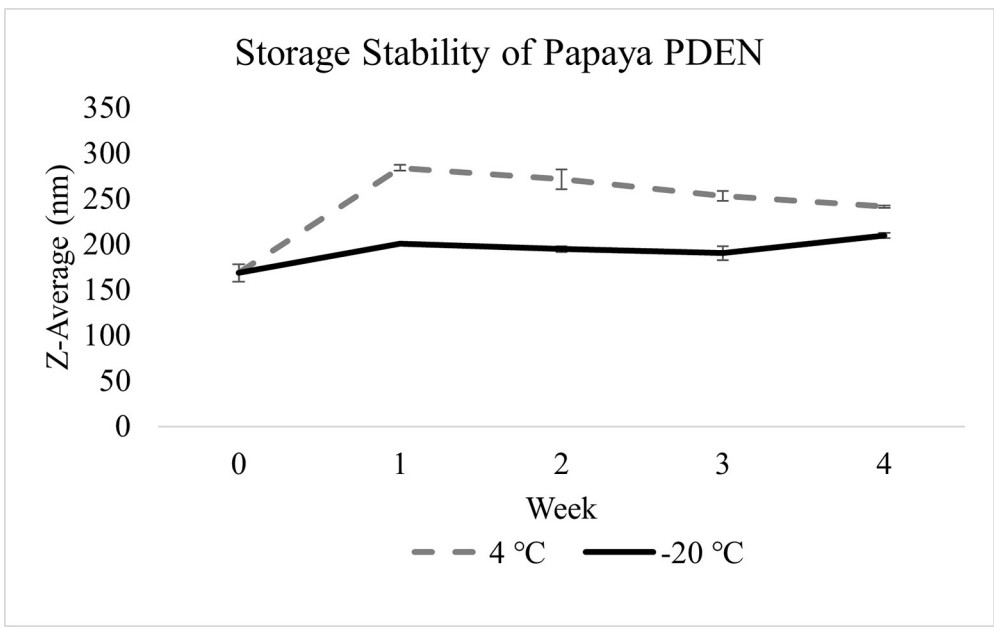

**Fig 4. Identification of the stability of papaya PDENs at different storage temperatures and weeks.** Papaya storage condition at 4˚C showed an increasing trend with a noticeable change in size in the first week. Meanwhile at -20˚C, the size of papaya PDENs only showed a slight increase in the second week, but it was relatively stable in the second to fourth week.

performed GC-MS for profiling bioactive compounds present in papaya PDENs. Typical GC-MS profiles are shown in S1 Fig. Organic compounds are the most abundant metabolite detected in papaya PDENs. We have found a high number of antioxidants in all samples, such as maltol, 4H-pyran-4-one-2,3-dihydro-3,5-dihydroxy-6-methyl, Methyl 5,12-Octadecadienoate, and phenol. Moreover, papaya PDENs contains abundant compounds which are shown to have a variety of pharmacological effects including anti-inflammatory, antifungal, antimicrobial, anticancer, and antiviral effects. The result showed distinct profile of metabolite among those periods of storage time (at temperature -20˚C). Nevertheless, each sample retains the pharmacological effects of the extracted bioactive compounds. It can also be concluded that -20˚C storage condition can maintain the quality of sample. Table 3 showed the presence of compounds extracted from several ages of papaya PDENs, its chemical structure, percentage areas of these compounds, and the respective biological activities reported in the consulted bibliography.

### Total polyphenolic content (TPC) estimation and antioxidant activity

According to recent findings, PDENs are a remarkable source of plant bioactive compounds in bioavailable form. These papaya PDENs have considerable antioxidant properties. Therefore, we further confirmed the presence of polyphenolics in papaya PDENS using biochemical methods. Total polyphenolic content (TPC) was measured using the Folin-Ciocalteu method as explained earlier [31] to the different ages of papaya PDENs. As represented in Fig 6A, the freshly isolated papaya PDENs contained about 500 ppm of TPCs. The result indicates a slight decrease in TPCs due to the period of storage. Even though, there are no significant differences in TPCs amongst those periods of storage (at -20˚C) statistically. Since total polyphenolic contents are highly correlated with antioxidant capacity, we also investigated the DPPH-free radical scavenging activity in papaya PDENs (Fig 6B). Freshly isolated papaya PDENs showed around 30% antioxidant activity. Statistically, there are significant differences between the

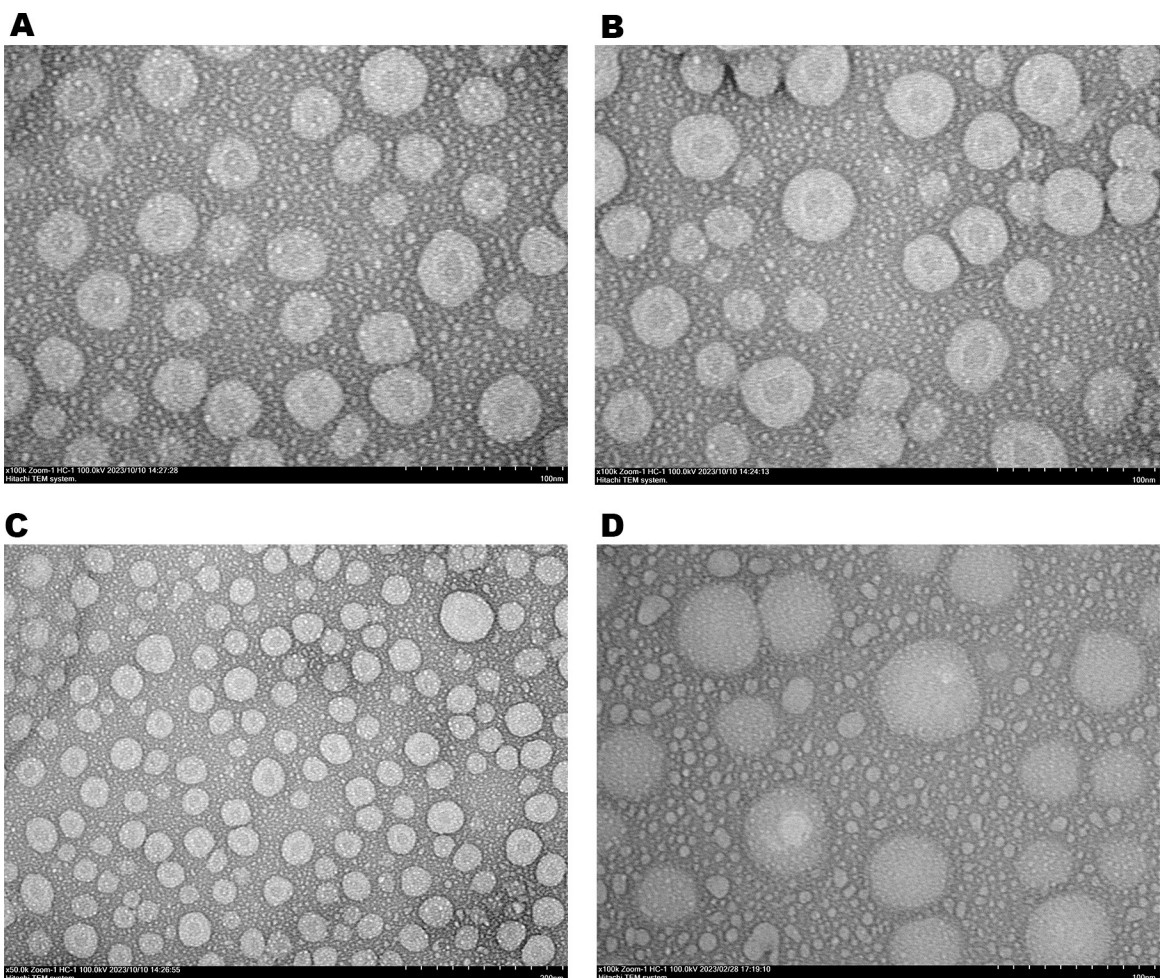

**Fig 5. TEM imaging of samples stored at -20˚C with different storage periods.** (A) 1 week, (B) 2 weeks, (C) 3 weeks, and (D) 4 weeks. The result showed the morphological structure of papaya PDENs was spherical and cup-shaped.

fresh and stored papaya PDENs with a decreasing trend of antioxidant activity. In the range of 2–4 weeks, the antioxidant activity remains stable with no significant difference. Therefore, this study focuses on the application of freshly isolated papaya PDENs for advanced biological analyses and assays.

## The uptake of papaya PDENs by RAW 264.7 cells

The cellular uptake of papaya PDENs was analyzed in RAW 264.7 cells through fluorescence observation using a confocal microscope after labeling PDEN with PKH67 (green). The results showed uptake of PDENs in RAW 264.7 cells coincubated with PKH67-labeled papaya PDENs (Fig 7). A continuous increase in cellular internalization was observed over a period of 2, 6, and 24 hours, where PKH67-labeled papaya PDENs was found surrounding the DAPI-stained nucleus (blue).

## RAW 264.7 cells viability

The cytotoxicity of papaya PDENs was assessed by exposing RAW 264.7 cells to varying concentrations of papaya PDENs (0, 5, 10, 20, and 100 μg/mL) for 24 hours, followed by the

**Table 3. Top-ranking metabolites identified in papaya PDENs age.**

| Sample | Compound | Area % | Biogical Action |
|---|---|---|---|
| A | .gamma.-sitosterol | 4,03 | Anticancer activity [40] |
| | 13-Docosenamide | 2,54 | Not reported |
| | 4H-pyran-4-one-2,3-dihydro-3,5-dihydroxy-6-methyl | 27,66 | antimicrobial, anti-inflammatory, and antioxidant capacity [41] |
| | 5-Hydroxymethylfurfural | 3,00 | Antioxidative, anti-allergic, anti-inflammatory, anti-hypoxic, anti-sickling, anti-hyperuricemic [42] |
| | 5-methylhexane-2,4-dione, keto form | 22,38 | Not reported |
| | Campesterol | 4,35 | Anti-inflammatory, Cancer prevention [40] |
| | Cycloserine | 1,31 | Antibiotic (inhibits bacteria cell-wall synthesis) [43] |
| | Hexadecanoid acid, 3-hydroxyl-1-(hydromethyl)ethyl ester | 2,01 | Antioxidant, antimicrobial [44] |
| | Maltol | 13,23 | Promotes mitophagy and oxidative stress inhibition [45] |
| | N-Hexadecanoic acid | 3,19 | Anti-inflammatory, anti-bacterial, anti-fungi, modulate immune response [44] |
| | Octane,3,4-dimethyl | 8,17 | Not reported |
| | Stigmasterol | 1,79 | Anti-inflammatory, immunomodulatory [40] |
| B | 10,13-Octadecadienoic acid | 11,70 | anti-inflammatory effect [44] |
| | 11-Octadecenoic acid | 20,84 | anti-inflammatory effect [44] |
| | 1-Decanamide | 1,10 | Antibacterial, antifungal [46] |
| | 1-Guanidinosuccinimide | 6,83 | Anti-tumor [47] |
| | 2-Ethylacridine | 6,36 | Not reported |
| | 9,12-Octadecadienoic acid | 19,84 | Anti-inflammatory effect [44] |
| | Ethyl starate | 5,40 | Apoptotic induction in tumor cell [48] |
| | Hexadecanoic acid | 15,63 | Anti-inflammatory, anti-bacterial, anti-fungi, modulate immune response [44] |
| | Methyl 5,12-Octadecadienoate | 3,09 | Antioxidant [49] |
| | Methyl 9-cis,11-trans-octadecadienoate | 3,95 | Antioxidant [49] |
| | Propanamide | 3,91 | Not reported |
| C | .gamma.-Sitosterol | 5,35 | Anticancer activity [40] |
| | 13-Docosenamide | 3,38 | Not reported |
| | 1H-indene, 5-butyl-6-hexyloctahydro | 1,31 | High antioxidant and anti-cancer [50] |
| | 4H-Pyran-4-one,2,3-dihydro-3,5-dihydroxy-6-methyl | 23,94 | antimicrobial, anti-inflammatory, and antioxidant capacity [41] |
| | 9,12,15-Octadecatrienoic acid | 1,76 | Anti-inflammatory effect [44] |
| | Butane,2-(ethenyloxy)-2-methyl | 18,44 | Not reported |
| | Campesterol | 4,04 | Cancer prevention [40] |
| | Cyclopentanone, dimethylhydrazone | 13,17 | Not reported |
| | Glycerol 1-palmitate | 2,00 | Anti-viral effect [51] |
| | Hexane-2,3,5-trimethyl | 9,56 | Not reported |
| | N-Hexadecanoic acid | 4,44 | Anti-inflammatory, anti-bacterial, anti-fungi, modulate immune response [44] |
| | Nonanal | 4,41 | Antifungal [52] |
| | Stigmaserol | 3,04 | Anti-inflammatory, immunomodulatory [40] |
| D | 2-Ethylacridine | 5,40 | Antitumor, antioxidant [53] |
| | 2-OXOPROPIONAMIDE | 27,52 | Anti-inflammatory, anti-fungal, anti-microbial [54] |
| | 4H-Pyran-4-one,2,23-dihydro-3,5-dihydroxy-6-methyl | 19,21 | antimicrobial, anti-inflammatory, and antioxidant capacity [41] |
| | Acetic acid | 1,59 | Antibacterial (penetrate into the cell membrane of microorganisms) [55] |
| | Glyceraldehyde | 2,83 | Not reported |
| | Hexadecanoic acid, methyl ester | 2,24 | Anti-inflammatory, anti-bacterial, anti-fungi, modulate immune response [44] |
| | Piperazine | 2,14 | Anti-inflammation, anti-cancer [56] |
| | Piperazine,1,4-dimethyl | 19,39 | Antifungal, antibacterial, antiviral, antioxidant, anticancer, anti-inflammatory, cognition enhancer [56] |
| | Pterin-6-carboxylic acid | 1,13 | Antioxidant, antimicrobial, anticancer [57] |
| | Valeraldehyde, dimethylhydrazone | 17,46 | Not reported |

*(Continued)*

**Table 3.** (Continued)

| Sample | Compound | Area % | Biogical Action |
|---|---|---|---|
| E | .gamma.-Sitosterol | 6,11 | Anticancer activity [40] |
| | 10,12-Hexadecadien-1-ol acetat | 2,34 | Anti-inflammatory, anti-bacterial, anti-fungi, modulate immune response [44] |
| | 3,4-Dihydroxyacetophenone (DHAP) | 1,19 | Prevent oxidative stress [58] |
| | 4-Benzofuranone,6,7-dihydro-3,6-dimethyl | 3,01 | Anti-tumor, antibacterial, antioxidative, and antiviral [59, 60] |
| | 4H-Pyran-4-one-2,3-dihydro-3,5-dihydroxy-6-methyl | 18,16 | Flavonoid; mutagen antimicrobial, anti-inflamatory, and antioxidant capacity [41] |
| | 6-Amino-1,3,5-triazine-2,4-dione | 4,04 | Anticancer [61] |
| | 9,12-Octadecadienoic acid | 11,24 | Anti-inflammatory effect [44] |
| | 9,17-Octadecadienal | 5,37 | Anti-inflammatory effect [44] |
| | Campesterol | 4,91 | Cancer prevention [40] |
| | Cyclohexene | 2,61 | Against Gram-positive bacteria [62] |
| | Ethanimidic acid | 3,26 | Not reported |
| | Hexadecanoic acid | 5,17 | Anti-inflammatory, anti-bacterial, anti-fungi, modulate immune response [44] |
| | Methyl 9,12-Octadecadienoate | 1,81 | Anti-inflammatory effect [44] |
| | Methyl starate | 1,35 | Apoptosis induction [48] |
| | Phenol | 1,89 | Antimicrobial, analgesic, anti-inflammatory, antioxidant, anti-cancer, antiseptic and disinfecticides [63] |
| | Propylcarbamate | 23,57 | Enhancing stability and pharmacokinetic [64] |

(A) 0 week (freshly isolated), (B) 1 week, (C) 2 weeks, (D) 3 weeks, and (E) 4 weeks.

administration of MTT reagent. The results indicate that at each concentration, papaya PDENs falls into the non-cytotoxic category (viability > 80%), with cell viability at concentrations of 5, 10, 20, and 100 μg/mL being 103.7%, 97.2%, 95.1%, and 97.3%, respectively (Fig 8).

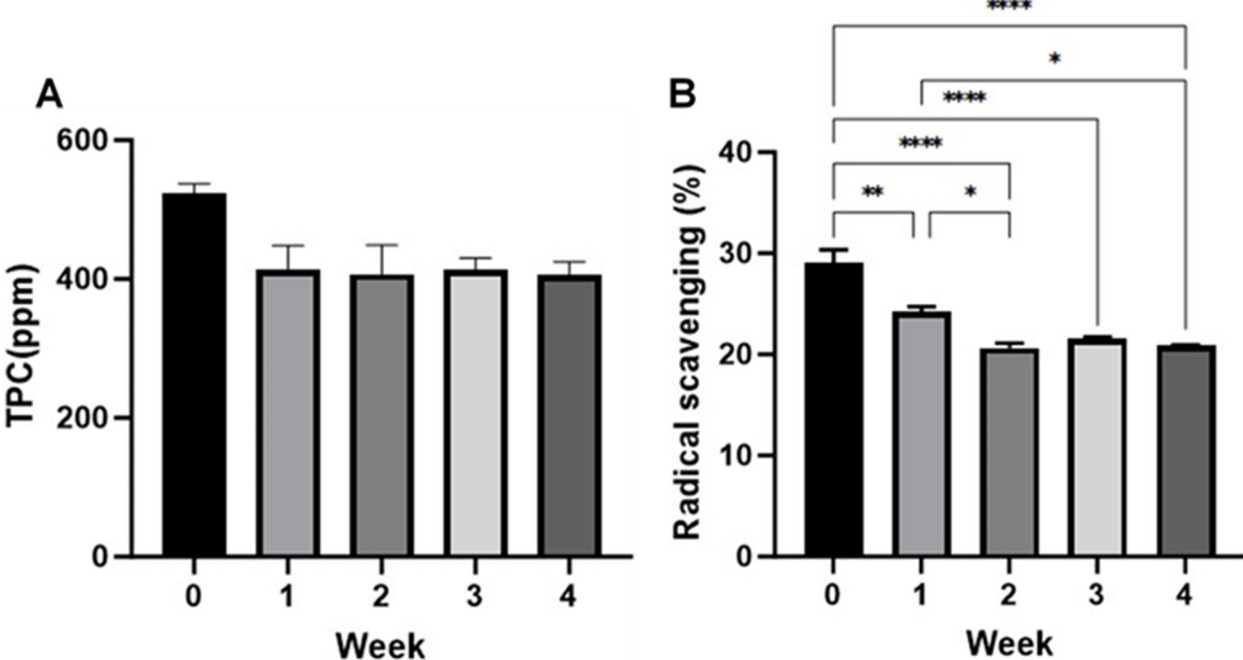

**Fig 6. Total polyphenols are present in papaya PDENs and show antioxidant activity, *in vitro*.** (A) Total polyphenolics (TPC) measured by Folin Ciocalteu methods toward five different ages of papaya PDENs. (B) Antioxidant activity of papaya PDENs assayed by DPPH methods. (* p < 0.05, ** p < 0.01, *** p < 0.001, **** p < 0.0001, no additional bar: not significant).

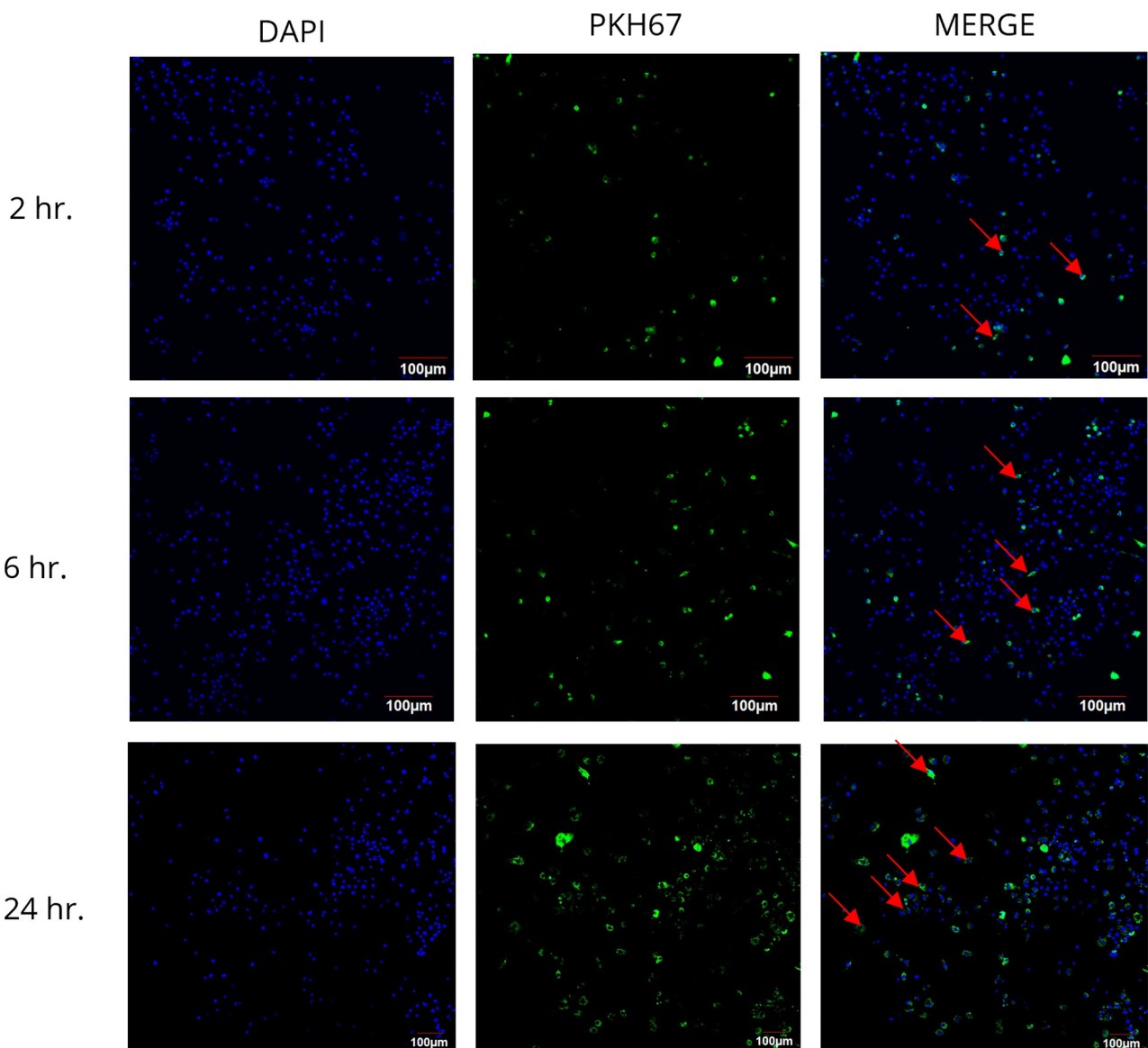

**Fig 7. Internalization of papaya PDENs in RAW 264.7 cells.** Internalization of PDENs shown using confocal microscope images. PKH67-labeled papaya PDENs (green) and DAPI-stained nucleus (blue).

## Nitric oxide assay

The nitric oxide (NO) concentration was analyzed in RAW 264.7 cells. The cells were cultured with different concentrations of PDENs (5, 10, and 20 μg/mL) and then stimulated with LPS (50 ng/mL). The nitrite concentration in the cell culture medium was measured to detect the increase in NO production due to LPS treatment. The aim of this study is to evaluate the ability of PDENs to inhibit NO production in RAW 264.7 cells. The study showed that LPS-induced RAW 264.7 cells treated with dexamethasone and papaya PDENs exhibited lower levels of nitric oxide compared to the LPS-induced RAW 264.7 cells (Fig 9). Therefore, dexamethasone and papaya PDENs were found to significantly reduce NO production in LPS-induced RAW 264.7 cells, but no significant differences were observed at any concentration of papaya PDENs treatment.

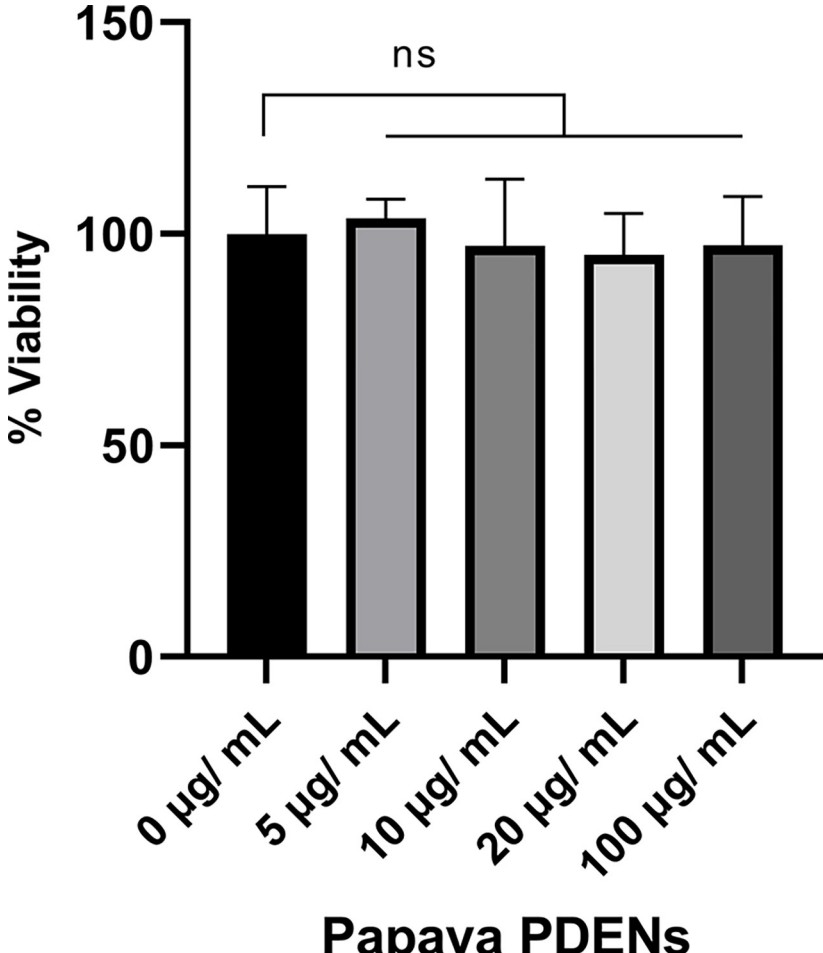

**Fig 8. MTT assay of RAW264.7 cell line treated with papaya PDENs for 24h.** Papaya PDENs demonstrated non-cytotoxic properties, as the cell viability percentage remained above 80% upon treatment with all concentrations of papaya PDENs. (ns: not significant).

### qRT-PCR

We analyzed the relative expression of inflammatory cytokine mRNA in RAW 264.7 cells by subjecting them to various concentrations of PDENs (5, 20, and 100 μg/mL) followed by stimulation with LPS (50 ng/mL). The aim was to assess the potential of PDENs to downregulate the expression of pro-inflammatory cytokines (IL-6 and IL-1β) and to upregulate the expression of anti-inflammatory cytokines (IL-10) in RAW 264.7 cells. We observed that pre-treatment with papaya PDENs significantly downregulated the relative expression of IL-6 mRNA in comparison to LPS-induced cells; however, it did not exhibit a dose-dependent manner (Fig 10A). Similarly, a significant downregulation of the relative expression of IL-1β mRNA was observed at a concentration of 20 μg/mL compared with LPS-induced cells, but not in a dose-dependent manner (Fig 10B). Additionally, relative expression of IL-10 mRNA showed significant upregulation when compared to LPS-stimulated cells, but it did not exhibit a dose-dependent manner (Fig 10C).

### Anti-inflammation effect of papaya PDENs on zebrafish

To study the cellular inflammatory response, zebrafish were immersed in varying concentrations of papaya PDENs (5, 20, 100 μg/mL) during the larval stage. Inflammation was induced

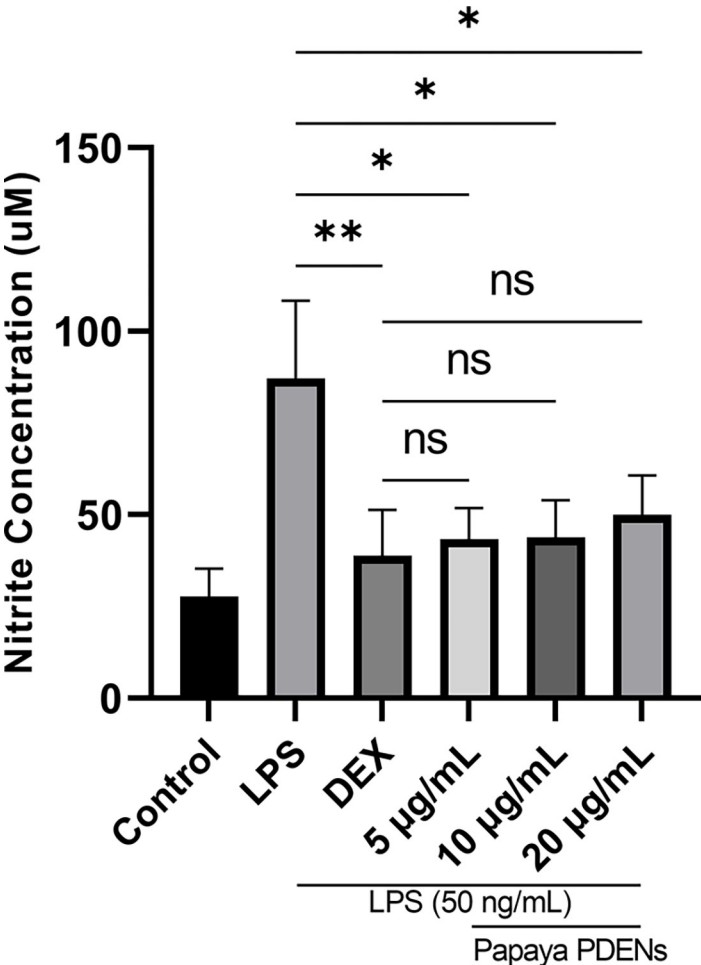

**Fig 9. Effect of papaya PDENs on the nitric oxide production in LPS-induced RAW264 cells.** Cell were treated with papaya PDENs for 24 h then treated with or without LPS (50 ng/mL) for 6 h (Control: No treatment RAW 264.7 cells, LPS: LPS-induced RAW 264.7 cells, DEX: Dexamethasone treatment) (* $p < 0.05$, ** $p < 0.01$, ns: not significant).

by the amputation of the caudal fin. Macrophages and neutrophils were visualized through neutral red and Sudan black staining, respectively. Live images of the caudal fin were taken with a stereo microscope four hours after the amputation of the fin. It was found that papaya PDENs significantly inhibited the migration of neutrophils to the wound site compared to control group and was not dose-dependent (Fig 11A). There was no significant difference observed between each PDENs treatment and the dexamethasone treatment (DEX). During the inflammatory response, a high number of neutrophils were observed in the control group. These cells appeared to align themselves along the edge of the wound site, in contrast to the PDENs treatment (Fig 12).

A similar effect was observed in macrophages, where papaya PDENs also suppressed macrophage migration to the wound site in comparison to control group (Fig 11B). This effect was not dose-dependent. At the papaya PDENs concentration of 5 µg/mL, a notable difference was observed when compared to control +. In addition, compared to the PDEN-treated group, there was a high accumulation of macrophages along the wound site in the control–(Fig 13). In addition, neutrophil and macrophage migration results were not significantly different from dexamethasone treatment at all papaya PDENs concentrations.

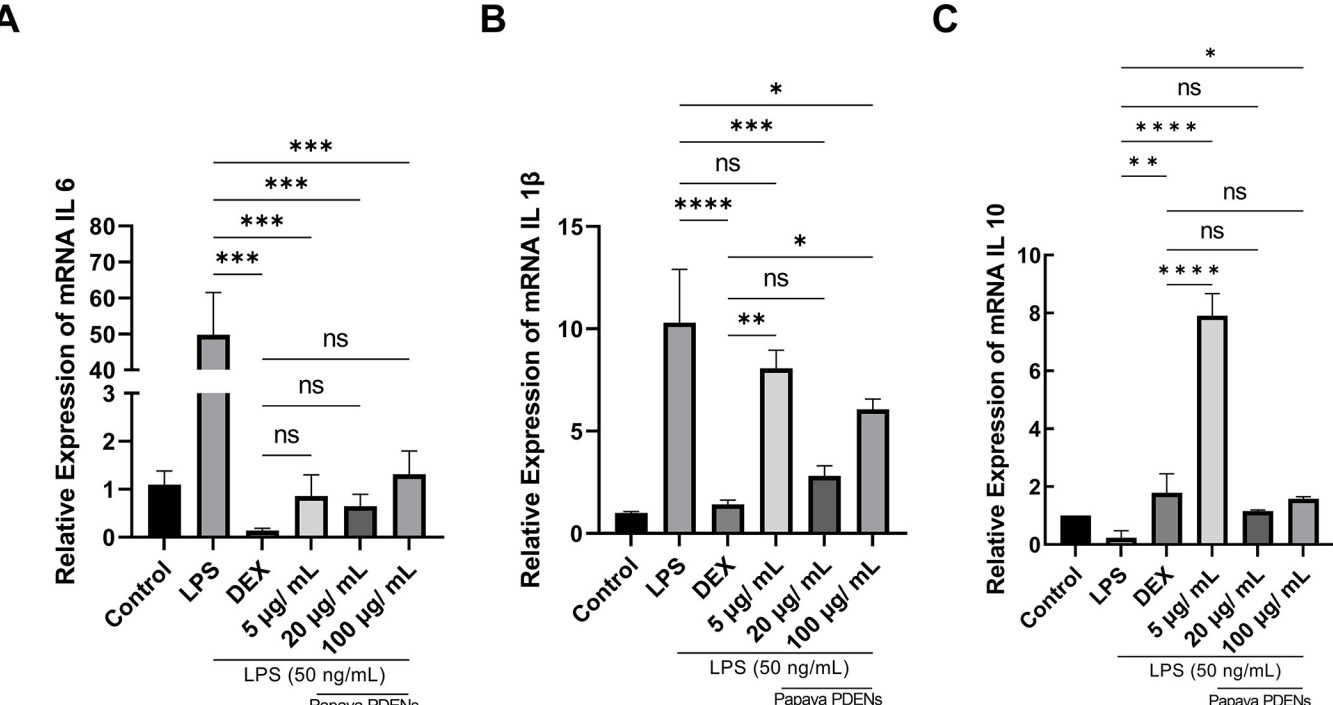

**Fig 10. Inflammatory cytokine relative expression on RAW 264.7 cells was analyzed by qRT-PCR.** (Control: No treatment RAW 264.7 cells, LPS: LPS-induced RAW 264.7 cells, DEX: Dexamethasone treatment). Papaya PDENs significantly downregulated the relative expression of IL-6 in all concentrations (A) and IL-1β from concentration 20 μg/mL (B) in comparison with LPS-induced RAW 264.7cells. It can also significantly upregulated relative expression of IL-10 (C) as compared to LPS-induced RAW 264.7 cells (* $p < 0.05$, ** $p < 0.01$, *** $p < 0.001$, **** $p < 0.0001$, ns: not significant).

## Discussion

In this study, PDENs was isolated from papaya fruit using differential centrifugation methods [31] with further optimization. The result showed that the best parameters for isolating papaya PDENs are the addition of 15% PEG, 0.2 M NaCl, and 30 ppm pectolyase, which generate spherical and cup-shaped PDENs with the average diameter size of 168.8 ± 9.62, a PDI of 0.261 ± 0.045, and a zeta potential value of 9.4 ± 0.2 mV. Papaya contains several bioactive secondary metabolites, including alkaloids, phenolics, flavonoids, carotenoids, tannins, and saponins, as well as proteolytic enzymes like papain and chymopapain. These substances possess anti-inflammatory, antimicrobial, immunomodulatory, and antioxidant activities [6, 17]. Papaya has a relatively high pectin content ranging from 0.66 to 2.03% [65], making it difficult to extract due to the gel-like formation. The addition of 30 ppm pectolyase was able to overcome this issue and produce isolated papaya PDENs with previously described characteristics and anti-inflammatory potential. Micronutrients, polyphenolic components, and antioxidants are bioactive compounds that enhance the values of papaya PDENs. Equally, the particle size is also important since larger nanoparticles result in a receptor shortage, which decreases cellular uptake because of the increasing entropic penalty [66]. In this study, the PEG-based method was used to isolate papaya PDENs, a cost-effective approach for PDENs isolation [31], with the addition of sodium chloride (NaCl) to improve the efficiency of collection [67].

Based on our study, it can be seen that the maturity of papaya fruit, the concentration of PEG6000, NaCl, and enzyme pectolyase affect the isolation efficiency of papaya PDENs. Mature fruit showed smaller particle size compared to the raw one. Enzymatic liquefaction in mashed papaya using 30 ppm of pectolyase exhibited the best result based on the viscosity

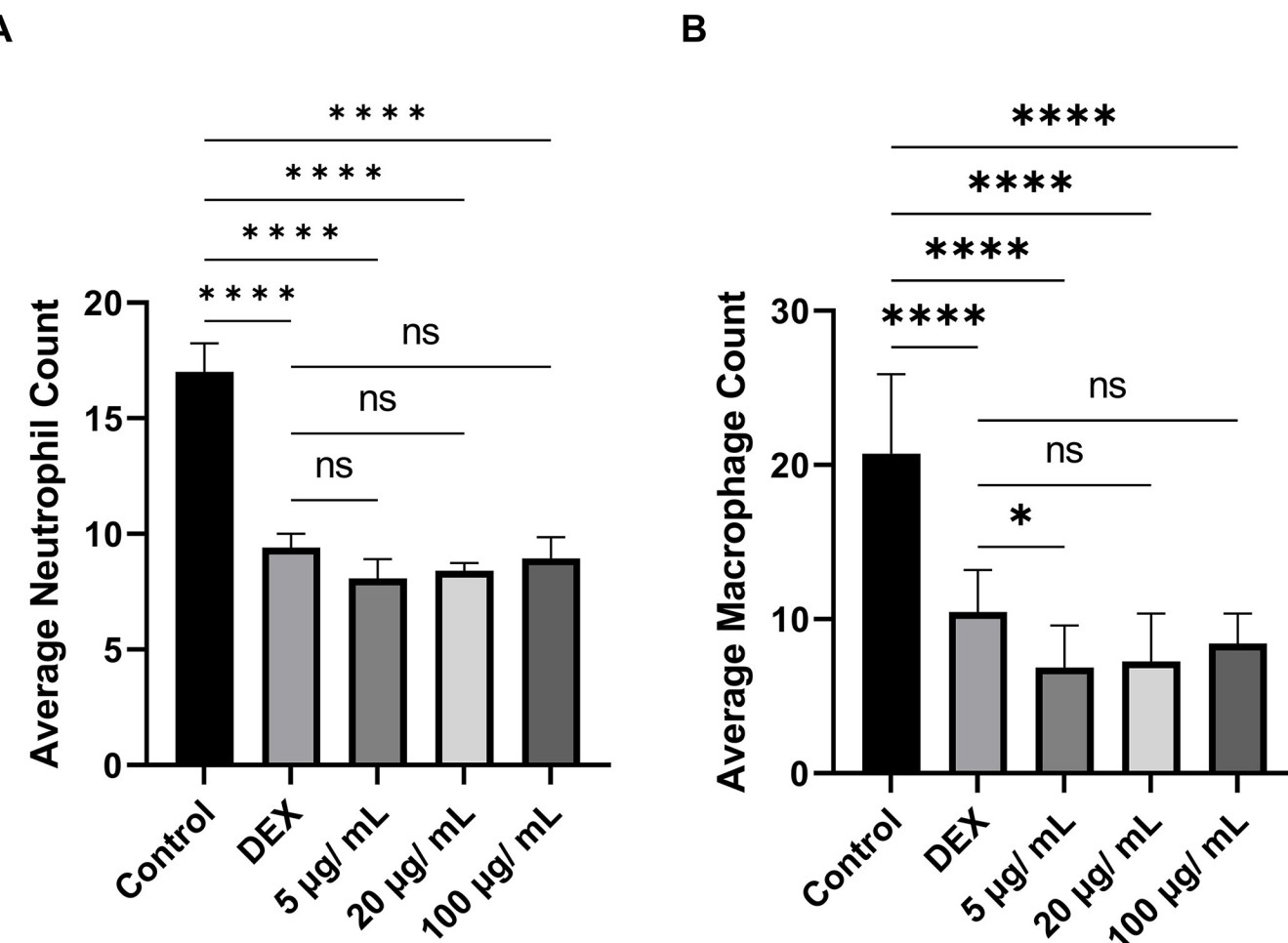

**Fig 11. Caudal fin amputation in zebrafish larvae as a model for inflammation.** (Control: No treatment wound-induced zebrafish larvae, DEX: Dexamethasone treatment). (A) Number of neutrophils recruited to the wound site at 4 hour post-amputation (4 hpa). (B) Number of macrophages recruited to the wound site at 4 hpa. (**** p < 0.0001, ns: not significant).

appearance, pellets obtained, and particle sizes. In this report, PEG6000 has been successfully used for the purification of papaya PDENs. As shown in Table 2, we observed an increase in papaya PDENs particle size with increasing concentration of PEG6000, as well as the PDI and kCPS value. The addition of 0.2 M sodium chloride indicates a notable improvement in the pellet collection efficiency. Our optimum isolation condition resulted in the size of papaya PDENs within the range of 50–500 nm [68] and a cup-shaped and spherical form [69], as expected. Additionally, the PDI value of PDENs papaya indicates a homogenous particle population suitable for drug delivery applications [70]. The zeta potential of PDENs papaya is negatively charged and exhibits limited stability [71]. The plasma membrane surface generally has a negative charge due to the presence of negatively charged glycosylated proteins that are integrated into the lipid bilayer membrane [72]. From each plant source, the PDENs will have a different set of characteristics. Some examples include ginger PDENs size 250–400 nm, spherical morphology, and negatively charged [31, 73, 74], apple PDENs 100–200 nm and spherical [75, 76], and grapefruit PDENs 105.7–396.1 nm and negatively charged [77].

Plant crude extracts and extracellular vesicles (EVs) are commonly stored at temperatures of -20˚C or -80˚C to preserve their biological activities [1]. PDENs papaya exhibits optimal

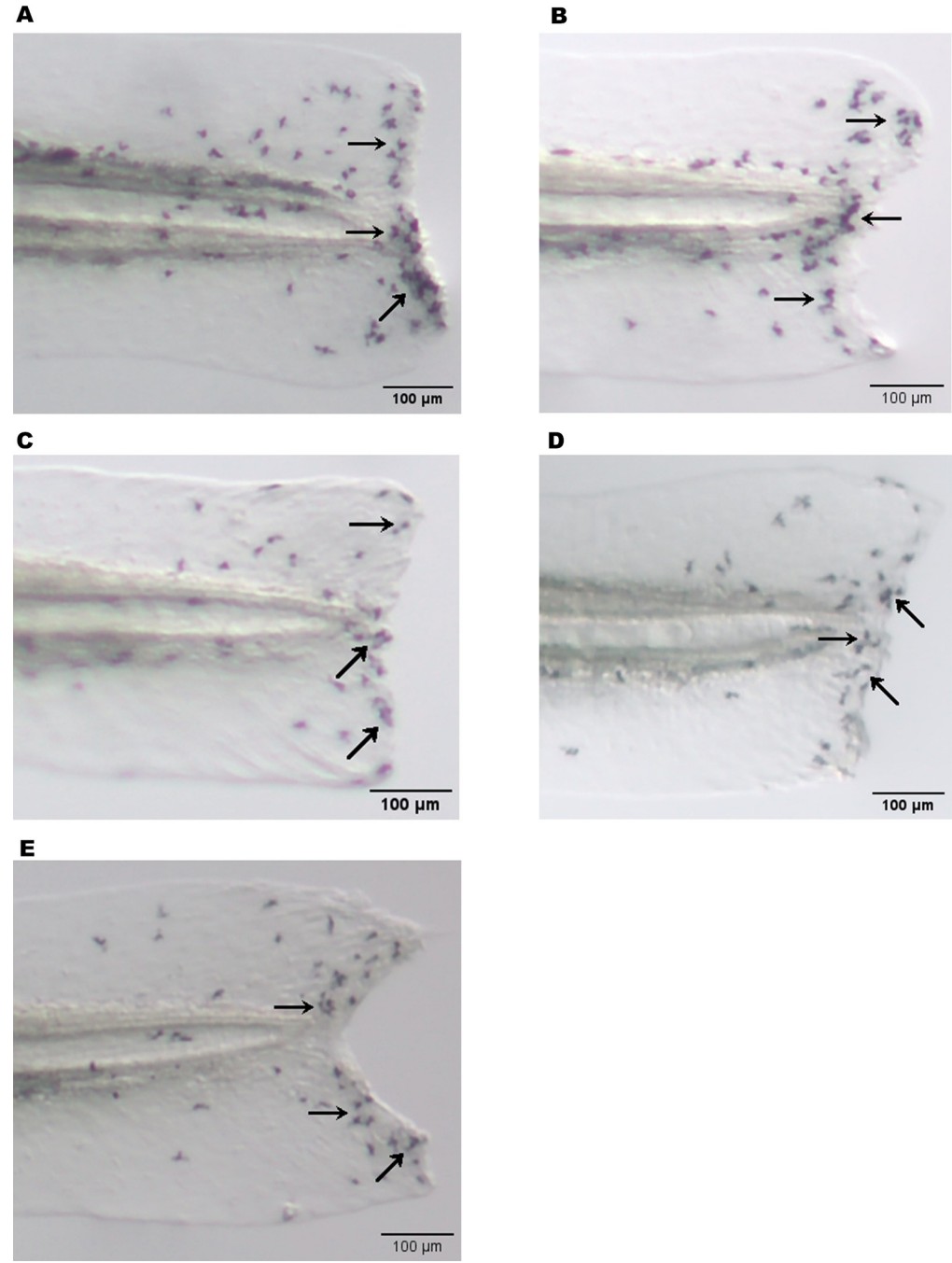

**Fig 12. Photograph of neutrophils in the amputated caudal fin of zebrafish larvae under stereo microscope.**
Neutrophils appear as black after staining with sudan black (black arrow). (A) PDEN-free treatment (control -). (B) Dexamethasone treatment. (C) 5 µg/mL. (D) 20 µg/mL. (E) 100 µg/mL.

physical stability in terms of particle size and dispersion at -20˚C. Trehalose is a frequently utilized cryoprotectant for EVs as it is capable of stabilizing proteins, cell membranes, and liposomes whilst also being safe. The objective of trehalose is to decrease the creation of ice crystals inside cells in freezing conditions through hydrogen bonding with water [78–80]. Trehalose-

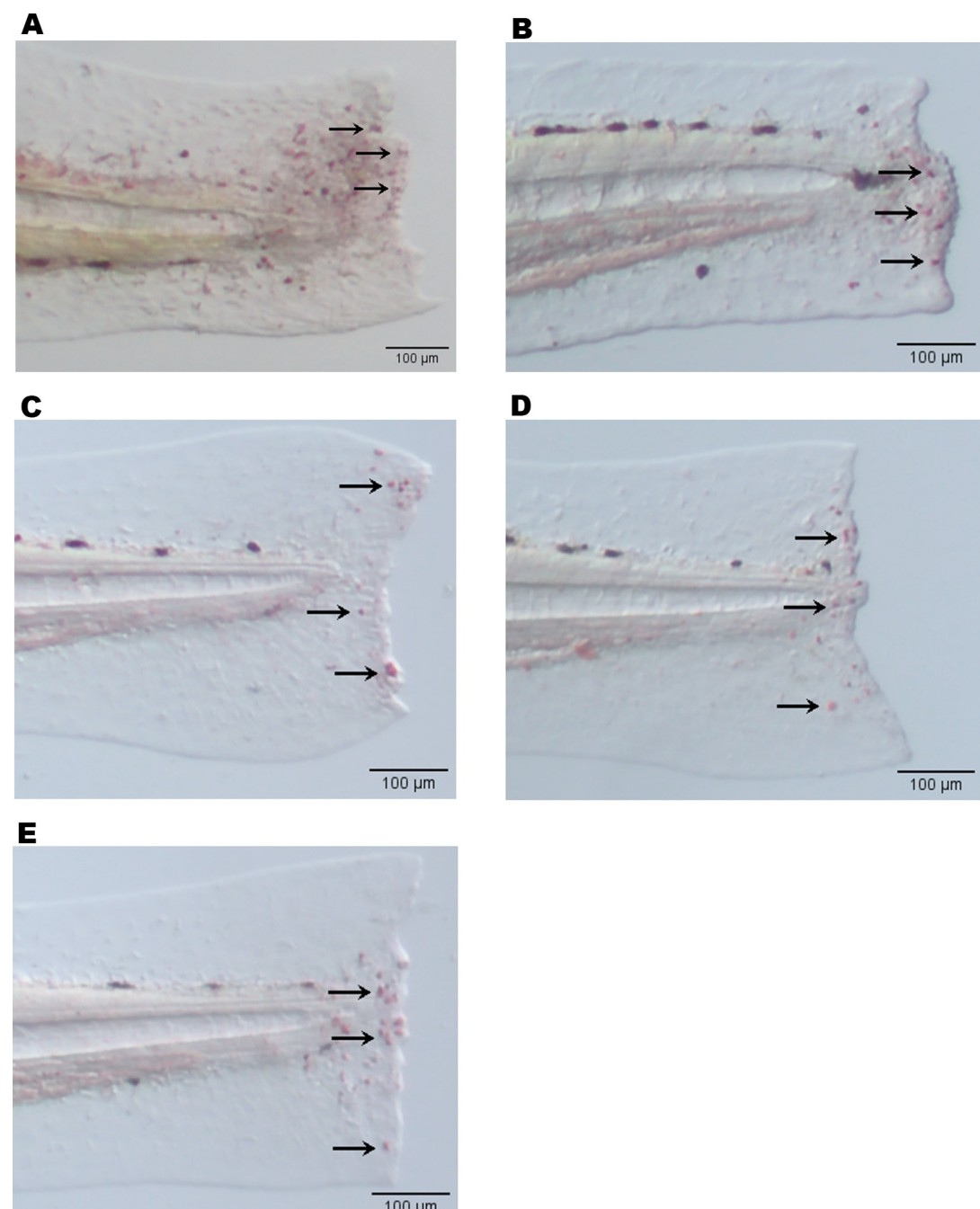

**Fig 13. Photograph of macrophages in the amputated caudal fin of zebrafish larvae under stereo microscope.** Macrophages appear as red after staining with neutral red (black arrow). (A) PDEN-free treatment (control). (B) Dexamethasone treatment. (C) 5 μg/mL. (D) 20 μg/mL. (E) 100 μg/mL.

treated PDENs have a smaller z-average diameter than untreated PDENs due to the presence of aggregates [81].

There have been other studies on the storage stability of PDENs, including the following: It was found that PDENs *Dedropanax morbifera* is best stored at -20˚C for 4 weeks [82]. Black ginger PDENs can remain effective after being stored at -20˚C and -80˚C for 8 weeks as long

as there is no repeat of the freeze-thaw cycle [32]. Another study suggests that *Aloe vera* PDENs can maintain its morphology for up to 90 days if stored at -20°C [83].

Papaya fruit contains various bioactive compound, including carotenoids, ascorbic acid, phenolics, saponins organic acid, and organic compound, which possess pharmacological effects [84]. Phytochemicals derived from papaya fruits exhibit various bioactivities such as anti-inflammatory, anti-oxidative, anti-cancer, anti-bacterial, anti-fungal, anti-viral, as well as modulating immune response [17, 85, 86]. In this study, by building on the proven beneficial health effect of papaya we have investigated the papaya PDENs isolated from papaya fruit. In spite of their promising therapeutic effect, their metabolite cargoes remain unknown. To understand the compounds responsible for bioactivity of PDENs, we analyzed the metabolome profiles of different ages of papaya PDENs. GC-MS study shows distinct profiles for freshly isolated, 1-, 2-, 3-, and 4-weeks papaya PDENs.

Most of these constituents have been found to exhibit interesting pharmacological properties. Further investigation identified 2,3-dihydro-3,5-dihydroxy-6-methyl-4H-pyran-4-one (DDMP), a flavonoid, as one of the major compounds in papaya PDENs which have shown a considerable therapeutic effect. Previous study reported that DDMP at a concentration 17.5 μM showed a good radical scavenging activity (81.1%), exceed butylated hydroxytoluene (58.4%) which is often used as a free radical scavenger [87]. Besides, DDMP is also reported as a promising anti-inflammatory, anti-bacterial, antifungal [88], and anti-cancer activities [89]. We also found maltol, Methyl 5,12-Octadecadienoate, Methyl 9-cis,11-trans-octadecadienoate, Pterin-6-carboxylic acid, 3,4-Dihydroxyacetophenone (DHAP), phenol, and 4-Benzofuranone,6,7-dihydro-3,6-dimethyl which also possess antioxidant capacity, as well as anti-microbial, anti-fungal, and anti-septic [45, 49, 57–60, 63]. These findings are also strengthened by the TPC and DPPH-assays result of papaya PDENs (Fig 6A & 6B).

Remarkably, we found considerable amount of anti-inflammatory agents in papaya PDENs with different action mechanisms, such as, N-hexadecanoic acid which is an inhibitor of phospholipase A(2) [90], 5-hydroxymethylfurfural as an inhibitor of MAPK, NF-κB, and Akt/mTOR pathways [91], stigmasterol which decreased NKR1-R expression in IL-13-induced BEAS-2B [92], 9,12-Octadecadienoic acid, 11-Octadecenoic acid, 10,13-Octadecadienoic acid [44], 2-OXOPROPIONAMIDE [54], and piperazine as histamine and serotonin receptor antagonists in the control of inflammation [93]. Our study represents a step forward in the analysis of metabolomics content of papaya PDENs isolated from papaya fruit. It exhibits that papaya PDENs can be a valuable source in pharmacology based on their capability to store bioactive compounds.

Although papaya PDENs originates from plants, our study has confirmed its internalization into RAW 264.7 cells. The way PDENs is internalized into mammalian cells remains unclear. Nonetheless, it is suspected that the internalization mechanism of PDENs is through the endocytosis pathway because there is endogenous phosphatidic acid on the surface of PDENs, which is generally associated with the internalization pathway of endocytosis. Phosphatidic acid functions as a secondary messenger that interacts with numerous proteins and regulates cell membrane transport systems. Furthermore, the endocytosis pathway typically mediates vesicle internalization in mammalian cells [32, 94, 95].

After investigating the physical properties of papaya PDENs, we tested its cytotoxicity on RAW 264.7 cells. PDENs papaya did not exhibit any significant cytotoxicity at any of the treatment concentrations. These findings aligned with other studies on PDENs extracts, such as Balloon flower root PDENs [96] and Lemon PDENs [97], where no significant cytotoxicity was observed.

Next, we examined the anti-inflammatory properties of papaya PDENs on LPS-activated RAW 264.7 cells. Inflammation significantly contributes to the complications of various

diseases, including diabetes, atherosclerosis, and cancer [15]. In this study, we found that papaya PDENs significantly inhibits the production of NO, which is consistent with previous studies conducted on PDENs group, such as Onion PDENs [5]. Activation of immune cells, specifically macrophages, could result in the production of inflammatory mediators including nitric oxide (NO) that could cause oxidative [98]. NO is formed after the interaction of reactive nitrogen species and reactive oxygen species (ROS) with toxic agents that are associated with inflammation [99]. Production of NO during the inflammatory process is often linked with iNOS [100]. Additionally, production of NO can result from the oxidation of L-arginine by inducible nitric oxide synthase (iNOS) and the conversion of arachidonic acid by COX-2. Thus, inflammation can be regulated by inhibiting NO production [99]. This current study suggests that papaya PDENs may potentially inhibit the action of iNOS and COX-2 enzymes, thus an anti-inflammatory agent.

We also found that papaya PDENs can downregulate the relative expression of pro-inflammatory cytokines such as IL-6 and IL-1β and also upregulate the relative expression of anti-inflammatory cytokines such as IL-10 in LPS-induced RAW 264.7 cells. These findings align with previous research on lemon PDENs, cabbage PDENs, balloon flower root PDENs, and ginger PDENs, all of which demonstrate inhibition of IL-6 and IL-1β cytokine expression and increased expression of IL-10, effectively suppressing inflammation in LPS-induced RAW 264.7 cells [33, 96, 97, 101].

However, the underlying mechanisms on the anti-inflammatory effects of papaya PDENs in vitro have not been investigated in this study. This study suggests that papaya PDENs, especially its bioactive compound, 5-hydroxymethylfurfural, could potentially inhibit the activation of the NF-κB pathway [91]. Furthermore, papaya PDENs also contains 2,3-dihydro-3,5-dihydroxy-6-methyl-4H-pyran-4-one (DDMP), which belongs to the flavonoid group and is known to have good anti-inflammatory properties [88]. Other polyphenolic natural product, such as the flavonol-enriched butanol fraction (UaB) from *Uvaria alba* have demonstrated its ability to inhibit the nuclear translocation of NF-κB p65 [102]. In addition, extracts from *U. alba* have been shown to inhibit human recombinant cAMP-specific phosphodiesterase (PDE4 B2). In the treatment of inflammatory diseases, PDE4 inhibitors are known to show potential [103]. Thus, papaya PDENs may have potential as an anti-inflammatory agent at mRNA level.

Furthermore, we evaluated the anti-inflammatory properties of papaya PDENs in zebrafish with caudal fin amputation. This amputation model has been widely used in testing anti-inflammatory agents [104, 105] and has resulted in the identification and early screening of various new anti-inflammatory agents [106, 107]. Our study showed that papaya PDENs effectively inhibited the migration of neutrophils and macrophages to the wound site. Our results are consistent with previous research on the zebrafish amputation model, which also observed a decrease in the migration of neutrophils and macrophages after the administration of ginsenoside extract to zebrafish [108]. Prolonged and excessive accumulation of immune cells can cause persistent inflammation. An effective anti-inflammatory agent will be able to suppress the migration of immune cells to the wound site to prevent chronic inflammation [109]. As an initial screening, the findings of this study suggest that papaya PDENs could have potential as an anti-inflammatory agent.

Neutrophils are the initial cells recruited to the site of tail injury [110]. Phosphoinositide 3-kinase (PI3K) facilitates cell migration by promoting actin polymerization and the formation of membrane protrusions at the leading edge through Rac activation and polarization of F-actin dynamics, which is required for actomyosin-mediated tail contraction [111]. NO is also a significant factor contributing to neutrophil migration [112], while macrophages migrate to wound site signaled by other chemoattractant molecules, such as hydrogen peroxide ($H_2O_2$)

[113]. Macrophage function has been shown to adapt depending on the presence of inflammatory molecules in their microenvironment (early wound signals, calcium, and reactive oxygen species/ROS) [114, 115]. While calcium triggers the recruitment and activation of M1 macrophages, ROS promote M1 polarization through NF-κB and Lyn pathways [115]. However, the underlying mechanisms on the anti-inflammatory effects of papaya PDENs in vitro have not been investigated in this study. We proposed that papaya PDENs may potentially decrease the production of NO, ROS, and $H_2O_2$. This leads to the suppression of leukocyte migration to the wound site and the inhibition of polarization into M1 macrophages, respectively [106, 107].

## Conclusions

In this study, our data showed that the best combination to isolate papaya PDENs are using mature papaya, 30 ppm pectolyase, 15% PEG, and 0.2 M NaCl, which exhibited PDENs size of 168.8 ± 9.62 nm, a zeta potential value of ~ -9.4 mV, and were stable when stored in aquabidest and 25 mM trehalose at a temperature of -20°C for up to one month. Our findings reported that papaya PDENs have remarkable bioactive compound such as antioxidant, anti-bacterial, anti-fungal, and anti-inflammatory effect. Moreover, RAW 264.7 cells are able to internalize these papaya PDENs. In vitro studies have shown that papaya PDENs did not induce cytotoxic effects, reduced nitrite concentrations, downregulated the expression of pro-inflammatory cytokines and upregulated the expression of anti-inflammatory cytokines. Additionally, the initial screening of potential anti-inflammatory agents in vivo using zebrafish demonstrated its efficacy in inhibiting the migration of neutrophils and macrophages to the wound site. As a result, papaya PDENs shows promise as a potential substitute for current inflammatory agents.

## Supporting information

**S1 Table. The Z-average, zeta potential, and PDI of the papaya PDENs stored at 4°C and −20°C for 0, 1, 2, and 4 weeks were analyzed by DLS.**
(PDF)

**S1 Fig. Metabolite profiles of papaya PDENs with different periods of storage.** (A) Freshly isolated, (B) 1 week, (C) 2 weeks, (D) 3 weeks, and (E) 4 weeks. Chromatograms show the most abundant GC-MS peaks.
(PDF)

## Acknowledgments

The authors wish to thank Dr. Indra Wibowo of the School of Life Sciences and Technology, Institut Teknologi Bandung who kindly provided zebrafish. We would also like to express our gratitude to IOBC (ITB Olympus Bioimaging Center) for providing confocal microscope.

## Author Contributions

**Writing – original draft:** Safira Vitasasti, Fatimah Nur Azmi Rahmadian.

**Writing – review & editing:** Iriawati Iriawati, Anggraini Barlian.

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
