## [Decision Letter · Decision Letter 0]

11 Jan 2024

PONE-D-23-43328Isolation and Characterization of Plant-Derived Exosome-Like Nanoparticles from Carica papaya L. Fruit and Their Potential as Anti-Inflammatory AgentPLOS ONE

Dear Dr. Barlian,

Thank you for submitting your manuscript to PLOS ONE. After careful consideration, we feel that it has merit but does not fully meet PLOS ONE’s publication criteria as it currently stands. Therefore, we invite you to submit a revised version of the manuscript that addresses the points raised during the review process.

We look forward to receiving your revised manuscript.

Kind regards,

Yong Sze Ong

Academic Editor

PLOS ONE

Journal Requirements:

-  https://doi.org/10.3390/cells9122722

In your revision ensure you cite all your sources (including your own works), and quote or rephrase any duplicated text outside the methods section. Further consideration is dependent on these concerns being addressed.

Reviewers' comments:

Reviewer's Responses to Questions

**Comments to the Author**

1. Is the manuscript technically sound, and do the data support the conclusions?

Reviewer #1: Partly

Reviewer #2: Yes

Reviewer #3: Yes

2. Has the statistical analysis been performed appropriately and rigorously? 

Reviewer #1: Yes

Reviewer #2: Yes

Reviewer #3: No

3. Have the authors made all data underlying the findings in their manuscript fully available?

Reviewer #1: Yes

Reviewer #2: Yes

Reviewer #3: No

4. Is the manuscript presented in an intelligible fashion and written in standard English?

Reviewer #1: Yes

Reviewer #2: Yes

Reviewer #3: Yes

5. Review Comments to the Author

Reviewer #1: This study explored the anti-inflammatory potential of papaya-derived exosome-like nanoparticles (PDENs). The researchers optimized a method for isolating PDENs and characterizing their morphology and stability. In vitro tests demonstrated internalization, non-cytotoxicity, and anti-inflammatory effects on RAW 264.7 cells, including reduced nitric oxide production and the downregulation of pro-inflammatory genes (IL-1B and IL-6) alongside upregulation of the anti-inflammatory gene (IL-10). In vivo tests on zebrafish showed the inhibition of macrophage and neutrophil migration. The researchers concluded that papaya PDENs exhibited promising anti-inflammatory properties, rendering them potentially valuable for therapeutic applications due to their safety and biocompatibility. Despite the intriguing findings, I recommend major revisions since the study lacks substantial data to showcase the anti-inflammatory effects of PDENs. To merit publication, the study should address the following:

1. Kindly cite examples of natural products with anti-inflammatory potential, specifically downregulating the NF-Kb pathway (DOI: 10.1021/acsomega.2c06451) and PDE4 B2B enzyme (DOI: 10.1021/acsomega.1c00137).

2. The quality of the figures is poor; kindly upload high-quality ones.

3. Table 4 identified the top-ranked metabolites present in the tested PDEN preparations. Given the current state of the paper, I believe it is not yet suitable for publication in PLOS One as it lacks rigor and in-depth analysis, especially in the biological assay part. To bridge this gap, I would require the authors to perform in silico analysis via molecular docking and computational pharmacokinetic evaluation, utilizing SWISS ADME and OSIRIS Toxicity software to further characterize the biological potential of the top-ranked compounds. Since the study hypothesized that the anti-inflammatory effects have something to do with the inhibition of the NF-Kb pathway, it is prudent to target the different enzymes and proteins from this pathway. For the computational analyses, please follow the protocols from these papers: DOI: 10.3390/computation10010007, DOI: 10.1080/07391102.2021.1969281.

4. At this point, there is no sufficient evidence to support the proposed mechanisms shown in Figure 12 even though they specifically mentioned it's only hypothetical, as no further assays were performed to specifically target NF-Kb and IL-10-related JAK-STAT machineries. I suggest excluding this figure.

5. Figure 11 should include tissue photomicrographs of the zebrafish used as the animal model.

6. The study is limited only to qRT-PCR data, which only shows mRNA expression. However, mRNA expression does not always correlate with protein expression. Please justify why western blot analysis was not carried out to investigate protein expression.

7. For the RT-PCR assay, kindly include the image of the gel to further showcase the effects of PDENs on the expression of target genes.

8. What are the limitations of the present study? Please provide recommendations to address these limitations.

Reviewer #2: The manuscript entitled “Isolation and Characterization of Plant-Derived Exosome-Like Nanoparticles from Carica papaya L. Fruit and Their Potential as Anti-Inflammatory Agent” used PEG Method to isolate papaya PDENs and characterized through DLS, TEM, BCA assay method, GC-MS analysis, total phenolic content (TPC) analysis, and 2,2-diphenyl-1-picrylhydrazyl (DPPH) assay. Also, the potential anti-Inflammatory effect were determined in vitro and vivo tests. The results indicated that papaya PDENs can be well isolated using the optimized differential centrifugation method with the addition of 30 ppm pectolyase, 15% PEG, and 0.2 M NaCl, which exhibited cup-shaped and spherical morphological structure with an average diameter of 168.8±9.62 nm. The papaya PDENs storage is stable in aquabidest and 25 mM trehalose solution at -20˚C until the fourth week. TPC estimation of all papaya PDENs ages did not show a significant change, while the DPPH test exhibited a significant change in the second week. The major compounds contained in Papaya PDENs is 2,3-dihydro-3,5-dihydroxy-6-methyl-4H-pyran-4-one (DDMP). The manuscript is interesting. In my opinion, the manuscript is well written and clear, but it has errors typographic, that is necessary minor corrections are required.

- Abstract

Please provide the full name of DLS, TEM, BCA.

-Key words

Please replace keywords that are in the title, basic rule.

- The manuscript needs another revision to check for spacing (spaces after words and before the citations are often missing).

Reviewer #3: 1. Kindly state the post-hoc test in the statistical analysis.

2. It is much appreciated if the images provided with the scale bar and measurement within the image.

3. Figure 1 and 2 somehow are blurred.

4. It is strongly recommend the images of zebrafish for the neutrophils and macrophages counts.

5. Kindly provide reference for line 78-79.

6. Typo at line 570, degree celcius double "C".

Overall, the manuscript is well written.

6. PLOS authors have the option to publish the peer review history of their article (what does this mean?). If published, this will include your full peer review and any attached files.

Reviewer #1: No

Reviewer #2: No

Reviewer #3: No

---

## [Author Response · Author response to Decision Letter 0]

23 Feb 2024

Dear Academic Editor and Reviewers,

We sincerely appreciate the valuable feedback and insightful comments provided by the academic editor and reviewers. We have carefully considered and addressed each of the reviewers' comments to ensure that the manuscript meets the publication standards and also support our original findings about the anti-inflammatory effects of PDENs from Carica papaya L. fruit. We hope the manuscript after careful revisions meet your high standards. Further elaboration on the modifications made can be found in the file "Respond to reviewers". 

Thank you for your consideration, and we eagerly await your response.

Kind regards,

Prof. Dr. Anggraini Barlian

Corresponding Author

---

## [Decision Letter · Decision Letter 1]

10 May 2024

Isolation and Characterization of Plant-Derived Exosome-Like Nanoparticles from Carica papaya L. Fruit and Their Potential as Anti-Inflammatory Agent

PONE-D-23-43328R1

Dear Dr. Barlian,

We’re pleased to inform you that your manuscript has been judged scientifically suitable for publication and will be formally accepted for publication once it meets all outstanding technical requirements.

Kind regards,

Piya Temviriyanukul, PhD

Academic Editor

PLOS ONE

Additional Editor Comments (optional):

The authors appropriately respond to the reviewers. Accept in its current form.

Reviewers' comments:

Reviewer's Responses to Questions

**Comments to the Author**

1. If the authors have adequately addressed your comments raised in a previous round of review and you feel that this manuscript is now acceptable for publication, you may indicate that here to bypass the “Comments to the Author” section, enter your conflict of interest statement in the “Confidential to Editor” section, and submit your "Accept" recommendation.

Reviewer #1: All comments have been addressed

Reviewer #2: All comments have been addressed

2. Is the manuscript technically sound, and do the data support the conclusions?

Reviewer #1: Yes

Reviewer #2: Yes

3. Has the statistical analysis been performed appropriately and rigorously? 

Reviewer #1: Yes

Reviewer #2: Yes

4. Have the authors made all data underlying the findings in their manuscript fully available?

Reviewer #1: Yes

Reviewer #2: Yes

5. Is the manuscript presented in an intelligible fashion and written in standard English?

Reviewer #1: Yes

Reviewer #2: Yes

6. Review Comments to the Author

Reviewer #1: The paper has been revised extensively and is now acceptable for publication. Congratulations to all authors!

Reviewer #2: The authors well respond to the comments. All issues are already clarified. In my opinion, this manuscript can be accepted in this version.

7. PLOS authors have the option to publish the peer review history of their article (what does this mean?). If published, this will include your full peer review and any attached files.

Reviewer #1: No

Reviewer #2: No

---

## [Editor Report · Acceptance letter]

2 Jun 2024

PONE-D-23-43328R1 

PLOS ONE

Dear Dr. Barlian, 

I'm pleased to inform you that your manuscript has been deemed suitable for publication in PLOS ONE. Congratulations! Your manuscript is now being handed over to our production team.

Kind regards, 

on behalf of

Dr. Piya Temviriyanukul 

Academic Editor

PLOS ONE